# Rft1 catalyzes lipid-linked oligosaccharide translocation across the ER membrane

Shuai Chen [1,2,3], Cai-Xia Pei [4,5], Si Xu[1], Hanjie Li[1], Yi-Shi liu[1], Yicheng Wang [2,3], Cheng Jin [4] ✉, Neta Dean [6] ✉ & Xiao-Dong Gao [1,2,3] ✉

The eukaryotic asparagine (N)-linked glycan is pre-assembled as a fourteen-sugar oligosaccharide on a lipid carrier in the endoplasmic reticulum (ER). Seven sugars are first added to dolichol pyrophosphate (PP-Dol) on the cytoplasmic face of the ER, generating Man5GlcNAc2-PP-Dol (M5GN2-PP-Dol). M5GN2-PP-Dol is then flipped across the bilayer into the lumen by an ER translocator. Genetic studies identified Rft1 as the M5GN2-PP-Dol flippase in vivo but are at odds with biochemical data suggesting Rft1 is dispensable for flipping in vitro. Thus, the question of whether Rft1 plays a direct or an indirect role during M5GN2-PP-Dol translocation has been controversial for over two decades. We describe a completely reconstituted in vitro assay for M5GN2-PP-Dol translocation and demonstrate that purified Rft1 catalyzes the translocation of M5GN2-PP-Dol across the lipid bilayer. These data, combined with in vitro results demonstrating substrate selectivity and *rft1Δ* phenotypes, confirm the molecular identity of Rft1 as the M5GN2-PP-Dol ER flippase.

Asparagine (N)-linked glycosylation is an essential posttranslational modification common to all three domains of life[1–4]. Before attachment to protein, the N-glycan is synthesized as a lipid-linked oligosaccharide (LLO). Most eukaryotes share a common fourteen sugar Glc3Man9GlcNAc2 (G3M9GN2, where Glc (G), Man (M) and GlcNAc (GN) are glucose, mannose and N-acetylglucosamine, respectively) precursor oligosaccharide whose stepwise assembly is catalyzed by the Alg (asparagine-linked glycosylation) glycosyltransferases[5,6]. These glycosyltransferase reactions are topologically distinct[7]. Seven sugars (GlcNAc-P, GlcNAc and five Man) are added to phosphate dolichol (P-Dol) on the cytoplasmic face of the ER[8–13]. The M5GN2-PP-Dol intermediate is then translocated across the ER membrane into the lumen[14] where another seven sugars (four Mans and three Glcs) are added[15–20] (Fig. 1a). Once assembled, the mature oligosaccharide is transferred *en bloc* to the asparagine residue of glycosylation sequences (Asn-Xaa-Ser/Thr) in nascent proteins by oligosaccharyl transferase (OST)[21,22].

There is compelling evidence that a specific protein transporter is required for flipping M5GN2-PP-Dol across the ER membrane[23–25]. This translocation is thermodynamically unfavorable because the polar glycan-head group must pass through the hydrophobic interior of the lipid bilayer. Moreover, this transbilayer translocation is rapid, ATP-independent, protease-sensitive, and occurs with exquisite substrate specificity[26], suggesting the existence of specific M5GN2-PP-Dol protein "flippase". Such a M5GN2-PP-Dol flippase was first reported by Helenius and Aebi twenty years ago. They identified Rft1 by yeast genetics as highly conserved and essential ER membrane protein required for the in vivo translocation of M5GN2-PP-Dol[27,28]. This interpretation was called into question by biochemical data showing that an in vitro flipping assay using proteoliposomes reconstituted with detergent-solubilized ER membrane proteins from rat liver or yeast cells, in which Rft1 was reduced or separated nevertheless displayed robust LLO-specific flippase activity[23,24,29]. A direct role of Rft1 was further rebuffed by the discovery that Rft1 is not essential for at least one eukaryote, *Trypanosoma brucei*[30]. To date, the identity of the flippase remains controversial and Rft1 remains the only protein implicated in either direct or indirect luminal translocation of M5GN2-PP-Dol.

[1]Key Laboratory of Carbohydrate Chemistry and Biotechnology, Ministry of Education, School of Biotechnology, Jiangnan University, Wuxi, China. [2]State Key Laboratory of Biochemical Engineering, Institute of Process Engineering, Chinese Academy of Sciences, Beijing, China. [3]Key Laboratory of Biopharmaceutical Preparation and Delivery, Chinese Academy of Sciences, Beijing, China. [4]State Key Laboratory of Mycology, Institute of Microbiology, Chinese Academy of Sciences, Beijing, China. [5]University of Chinese Academy of Sciences, Beijing, China. [6]Department of Biochemistry and Cell Biology, Stony Brook University, Stony Brook, New York, USA. ✉e-mail: jinc@im.ac.cn; neta.dean@stonybrook.edu; xdgao@ipe.ac.cn

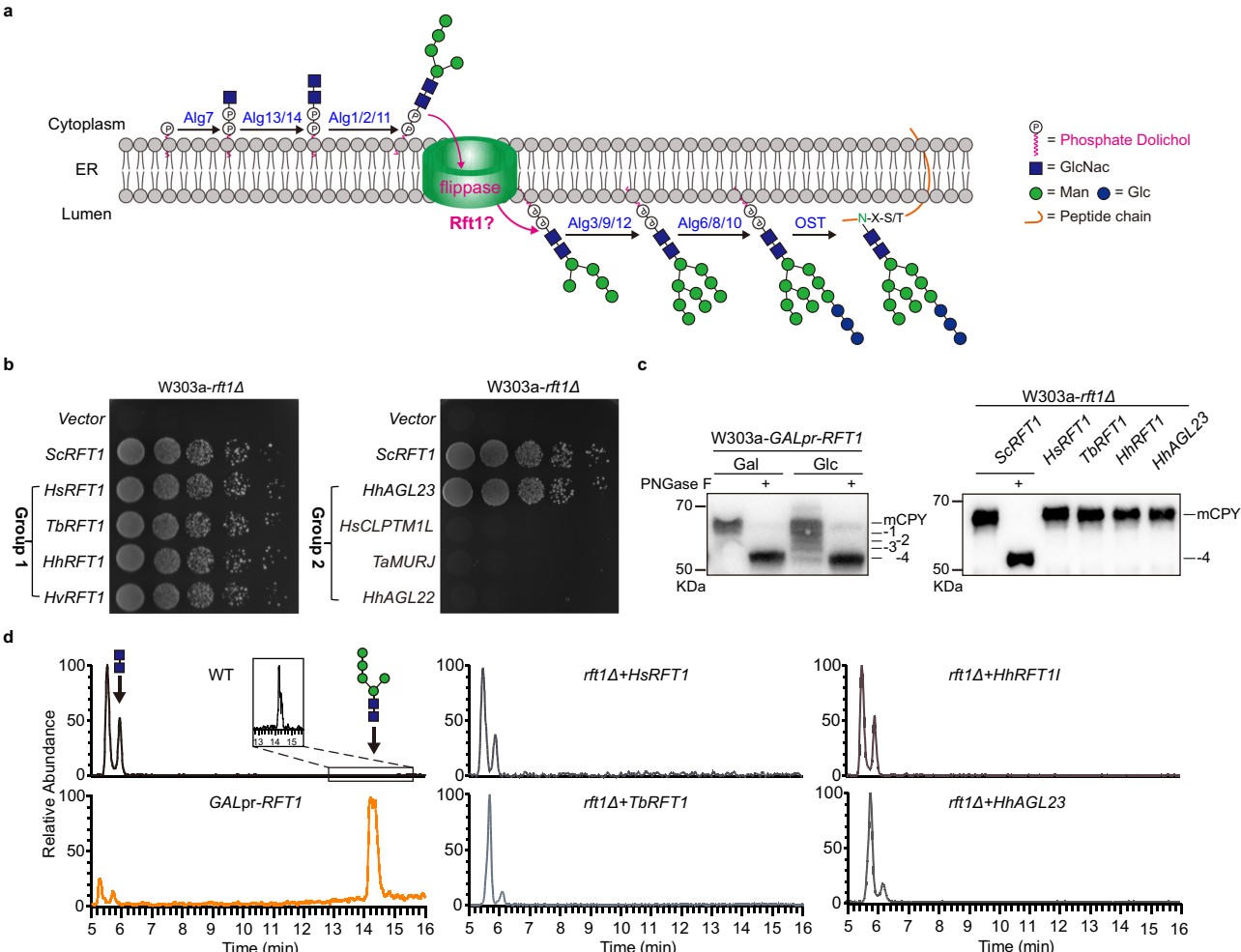

**Fig. 1 | High copy suppressors of *rft1Δ*. a** Schematic of ER lipid-linked oligo-saccharide biosynthesis of Man5GlcNAc2 (M5GN2)-PP-Dol in vivo, with the luminal translocation of M5GN2-PP-Dol in magenta. **b** Complementation of the *rft1Δ* growth defect. YEp351-GAPII plasmids, containing *ScRFT1*, *HsRFT1*, *TbRFT1*, *HhRFT1*, *HvRFT1*, *HhAGL23*, *HsCLPTM1L*, *TaMURJ*, *HhAGL22* or empty vector, were intro-duced in W303a-*rft1Δ* mutants containing pRS316-*ScRFT1*. Serial dilutions of each strain were plated on SD-Leu containing 5-FOA and incubated for 2 days at 30 °C. **c** Western blot analysis of CPY. *left*: Lysates were prepared from the glucose (Glc)-repressible *GALpr-RFT1* strain grown in galactose (Gal)- or Glc for 12 h at 30 °C and analyzed by SDS-PAGE followed by western blotting with anti-CPY antibodies; *right*: CPY in lysates from *rft1Δ* harboring high-copy Group I or Group II suppressors and was analyzed by Western blotting. Fully glycosylated (mCPY) and hypoglycosylated (lacking up to four oligosaccharides) CPY are indicated. Images are representative of two independent experiments. **d** UPLC chromatograms of oligosaccharides released from the LLO intermediates in different strains corresponding to those in **b** and in *GALpr-RFT1* cells at 12 h after shift to glucose-containing medium. Major peaks of GlcNAc2 and Man5GlcNAc2 are marked. Man and GlcNAc are indicated by circles (green) and squares (blue), respectively.

In addition to its strong conservation among eukaryotes, Rft1 orthologues are found in all major archaeal genomes[3]. Like eukaryotes, archaea pre-assemble N-glycans on dolichol[3,4]. As an alternate approach for characterization of the eukaryotic M5GN2-PP-Dol flip-pase, we turned to archaea since strong evidence supports the evo-lutionary conservation of the eukaryotic N-glycosylation pathway with archaeal enzymes[2–4].

In this study, with the idea that an ancestral archaeal flippase may reveal information about its eukaryotic orthologue, we searched for archaeal high copy suppressors of the lethality of a yeast *rft1* deletion strain. This approach led to the discovery that HhAgl23, an archaeal protein with no sequence resemblance to Rft1, com-plemented both the lethality and N-glycosylation defect of *rft1Δ* cells. Furthermore, we developed an in vitro LLO flippase assay comprised of fully reconstituted proteoliposomes and confirmed M5GN2-PP-Dol flipping activity from purified Rft1 or HhAgl23 protein. Our results demonstrated that Rft1 is directly responsible for M5GN2-PP-Dol translocation across the ER membrane during LLO biosynthesis.

## Results

### Identification of archaeal *rft1Δ* suppressors

To identify genes that suppress *rft1Δ* lethality, we used a plasmid shuffle approach. Candidates on a *2μ/LEU2* vector were transformed into a yeast *rft1Δ* strain, whose viability relies on the presence of a *URA3*-marked plasmid-borne copy of yeast *RFT1* (*ScRFT1*) (see Meth-ods). When cultured on media containing 5-FOA to force loss of the *URA3*-*ScRFT1* plasmid, only transformants with functional suppressors on the *LEU2* plasmid would allow *rft1Δ* growth. Candidate genes were classified into two groups: Group I encode proteins sharing structural homology with ScRft1, which were identified through BLAST[3]; Group II encode proteins having potential glycolipid flipping activity, and were identified from the literature.

As predicted based on sequence similarity, Rft1 orthologues from humans, trypanosomes, as well as two different species of halophilic euryarchaeota complemented the growth defect of *rft1Δ* [these included *HsRFT1* (*Homo sapiens*), *TbRFT1* (*Trypanosoma brucei*), *HhRFT1* (*Haloarcula hispanica*) and *HvRFT1* (*Haloferax volcanii*)]

(Fig. 1b). Group II candidates included human *CLPTM1L* (*HsCLPTM1L*), which translocates GlcN-PI and phospholipids across the ER[31], bacterial *Thermosipho africanus MurJ* (*TaMurJ*), which translocates lipid-linked peptidoglycan precursors[32] and archaeal *H. hispanica* Agl23 (HhAgl23), a putative flippase responsible for flipping of hexose-P-Dol[33]. Among these Group II, only HhAgl23 could suppress *rft1Δ* (Fig. 1b). Another *H. hispanica* glycosyltransferase in the N-glycosylation pathway HhAlg22[33], failed to suppress *rft1Δ*, suggesting HhAgl23 is a bona-fide functional suppressor of yeast *rft1Δ* (Fig. 1b).

To determine if *rft1Δ* suppression is due to a rescue of the N-glycosylation defect or to some other bypass mechanism, N-glycosylation of carboxypeptidase Y (CPY) in these strains was assayed. CPY is modified by the addition of four N-linked glycans[34]. Mutants defective in the synthesis, translocation or transfer of the core oligosaccharide display a characteristic ladder of CPY intermediates that contain 0,1,2,3 or 4 of these glycans. To assay CPY, lysates from various strains were analyzed by western forms were easily distinguished in a strain carrying *RFT1* under control of the glucose (Glc)-repressible *GAL1-10* promoter (*GALpr-RFT1* strain, see Supplementary Table 1 for description) when grown in galactose (Gal) or in Glc[28] (Fig. 1c, left panel). In *rft1Δ* cells harboring *RFT1* orthologues or *HhAGL23*, CPY migrated with a mobility similar to the fully mature form in the wild type (Fig. 1c, right panel).

To compare the levels of LLO intermediates, particularly M5Gn2-PP-Dol, in *GALpr-RFT1* and *rft1Δ* strains as a function of the various suppressor genes they harbor, an ultra-performance liquid chromatography-mass spectrometry (UPLC-MS) analysis was performed. It has been previously established that Rft1-depleted yeast strains accumulate the M5GN2-PP-Dol intermediate relative to wild-type cells[28]. By UPLC-MS, a large amount of the M5GN2 intermediate was observed in *GAL*pr-*RFT1* cells, compared to the wild type cells, at 12 h of incubation after the shift to glc-containing medium (Fig. 1d and Supplementary Fig. 1), thus confirming the M5GN2 accumulation phenotype of the Rft1-depleted strain. In *rft1Δ* cells, this accumulation was eliminated by expression of *RFT1* orthologues or *HhAGL23*. It is noteworthy that we also observed a large amount of GN2 intermediate in wild-type cells, and the relative amount of GN2 decreased concomitantly with M5GN2 accumulation in the glc-repressed *GALpr-RFT1* strain (Fig. 1d). An accumulation of GN2 intermediate in wild type cells was not reported previously, likely because previous methods used for measuring LLOs rely on isotopic labeling with [3H]-mannose. Thus, only mannose-containing LLOs can be labeled and detected. Taken together, our CPY mobility and UPLC-MS -based LLO analyses confirmed that suppression of *rft1Δ* lethality by Rft1 orthologues and by HhAgl23 is accompanied by a rescue of the N-glycosylation defect.

## Purified Rft1 and HhAgl23 translocate lipid-linked M5GN2 intermediate across the lipid bilayer

Suppression of *rft1Δ* by the *RFT1* orthologues was not surprising since they share structural homology with yeast Rft1. However, suppression by HhAgl23 was unexpected since it does not resemble Rft1 (Supplementary Fig. 2)[33]. Thus, we were interested in understanding how and why HhAgl23 is as efficient as Rft1 in its ability to complement both the lethality and glycosylation defect of *rft1Δ* (Fig. 1). *H. hispanica* Agl23 is a putative plasma membrane flippase for glc- or gal-P-Dol that are used as sugar donors for extracellular N- and O-glycan synthesis[33]. We hypothesized that when over-expressed in yeast, HhAgl23 functions as a flippase to translocate M5GN2-PP-Dol across the ER membrane in *rft1Δ*.

To test this idea and investigate the requirements of HhAgl23 and Rft1 during LLO translocation, we established an in vitro assay for detecting the M5GN2-PP-Dol flipping activity in the presence or absence of HhAgl23 or ScRft1 (shown schematically in Fig. 2a). Liposomes were generated with egg phosphatidyl-choline (PC) and brain phosphatidyl-serine (PS)[35,36] and contained a synthetic phytanyl

pyrophosphate M5GN2 (M5GN2-PP-Phy) substrate (see Fig. 2b for structure; Supplementary Fig. 3 for preparation). In these liposomes, M5GN2-PP-Phy adopted a random distribution between the inner and outer membrane leaflets, where about half of the oligosaccharide faces the lumen while the other half faces outside. This topological arrangement was determined by adding a purified α1-2 mannosidase to the reaction, to remove the two α1,2 linked mannose residues from M5GN2 in the outer leaflet (see Methods). The rational for this assay is that these external α1,2 linked mannoses should be accessible to mannosidase cleavage, while the lumenal mannoses are inaccessible and thus protected from cleavage (Fig. 2a and Supplementary Fig. 4). UPLC-MS analysis of liposomes in the presence or absence of α1-2 mannosidase confirmed that this treatment resulted in the predicted conversion of M5GN2 to M3GN2, with M3GN2 accounting for 50% of the oligosaccharide. As this was the maximum level of M5GN2 to M3GN2 conversion under these conditions, this experiment also demonstrated the absence of cross-leaflet M5GN2-PP-Phy exchange in these liposomes (Fig. 2a, upper; Supplementary Fig. 4). Using this assay, we predicted that the addition of a flippase to these liposomes would catalyze the exchange of M5GN2 between leaflets and therefore cause a greater conversion of M5GN2 to M3GN2 (up to 100%) due to the concomitant increased exposure of M5GN2 to α1-2 mannosidase (Fig. 2a, lower).

To determine if Rft1 or HhAgl23 catalyze M5GN2-PP-Phy translocation, we sought to reconstitute liposomes with purified ScRft1 or HhAgl23 in reactions that contained α1-2 mannosidase, without ATP (Fig. 2). After its purification (as described in Materials and Methods), ScRft1 migrated as single band on SDS-PAGE (Fig. 2c). To demonstrate that the single protein band was indeed ScRft1, it was extracted from the SDS-PAGE gel, subjected to proteolysis, and analyzed by LC-MS/MS. Sequence validation demonstrated ScRft1 identity and purity (Supplementary Fig. 5). HhAgl23 was similarly purified (Fig. 2c). These purified proteins were used in reconstituted liposomes and aliquots of the translocation reactions were analyzed by UPLC-MS. The results of this experiment indicated that with longer incubation times, decreased ratios of M5GN2 to M3GN2 were observed from proteoliposomes containing ScRft1 or HhAgl23. A total of 75% of M5GN2 was converted to M3GN2 after 2 h and 90% at 4 h (Fig. 2d). Protein-free liposomes or proteoliposomes containing a non-related protein purified in parallel (*H. hispanica,* HhAgl22) maintained a stable ratio of M5GN2 to M3GN2 of approximately 50% from the 1 h time point, which vanished after adding Triton X-100 at 4 h (Fig. 2d). To test if M5GN2-PP-Phy flipping by ScRft1 is dose-dependent, we measured end-point (2 h) flipping activity of Rft1-proteoliposomes prepared with different protein to phospholipid ratios. This experiment demonstrated the increased M5GN2 to M3GN2 conversion as a function of protein/phospholipid ratio (PPR), with M5GN2 decreasing to a plateau of approximately 25% in liposomes with a PPR of ~2 mg/mmol of purified ScRft1 (Fig. 2e). It should be noted that two different negative controls were included for this in vitro assay; protein-free liposomes and HhAgl22-containing proteoliposomes. As shown in Fig. 2d, HhAgl22-reconstituted proteoliposomes yielded a half-time of M5GN2 hydrolysis by α1-2 mannosidase that was similar to flippase-containing proteoliposomes. In contrast, a slight delay in hydrolysis was observed in protein-free liposomes. Nevertheless, taken together, these experiments demonstrated that both Rft1 and the archaea HhAgl23 can catalyze M5GN2-PP-Phy translocation in vitro.

To rule out the possibility that Rft1 flipping activity was an artifact associated with the synthetic PP-Phy lipid carrier used, we also assayed M5GN2 linked to dolichol, its physiological lipid. M5GN2-PP-Dol was synthesized[37] for use as a substrate (Fig. 2b and Supplementary Fig. 3). In proteoliposomes containing M5GN2-PP-Dol and Rft1, after 2 h of incubation, M5GN2-PP-Dol decreased to ~20%, a level similar to that using M5GN2-PP-Phy (Fig. 2f). In addition to yeast Rft1 (FLAG-ScRft1), recombinant human Rft1 (HsRft1-FLAG) was also tested (Fig. 2c, f). As

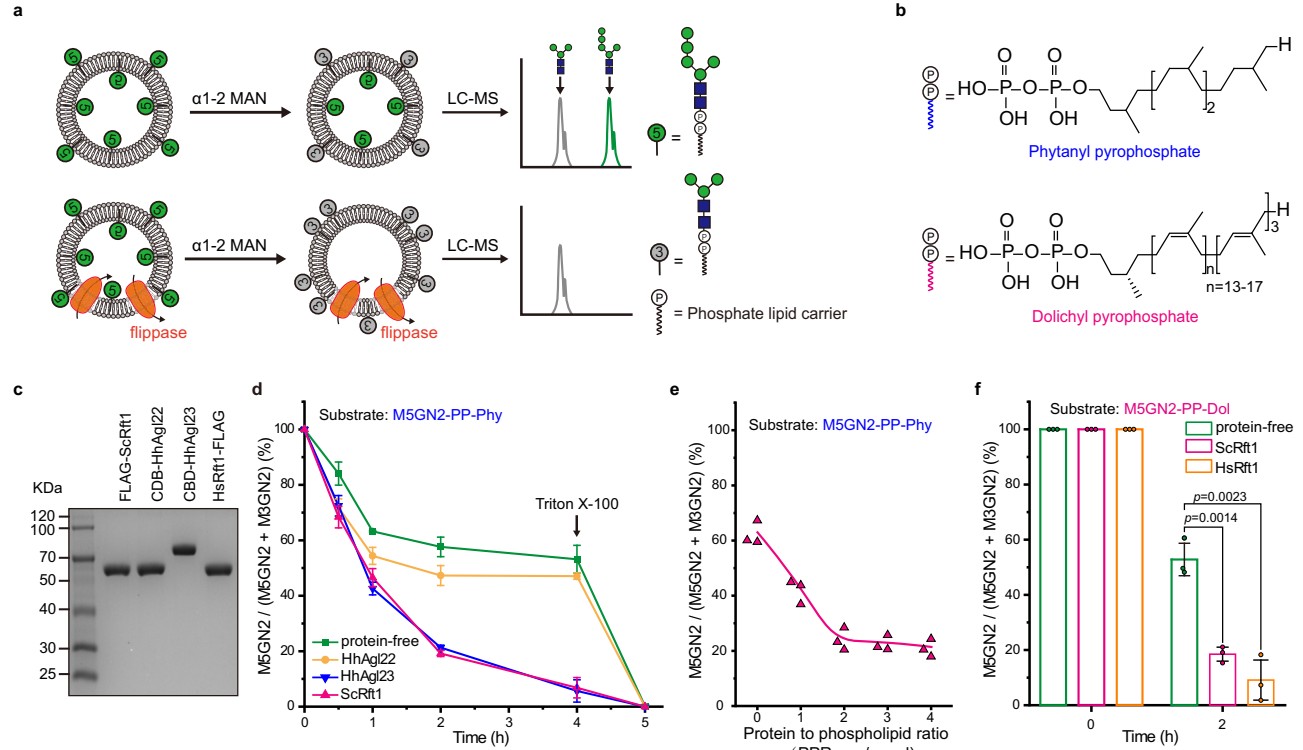

**Fig. 2 | Rft1 or HhAgl23 catalyze the translocation of M5GN2-PP-Pyh and M5GN2-PP-Dol across a lipid bilayer. a** Schematic of α1-2 mannosidase-based translocation assay by UPLC-MS analysis. Lipid-linked Man5GlcNAc2 (M5GN2) in the outer leaflet of liposomes is digested to Man3GlcNAc2 (M3GN2) by α1-2 mannosidase. In the absence of a flippase, there is a 1:1 distribution of M3GN2:M5GN2. Flippase activity is measured as an increase in this ratio. **b** Diagram of the phytanyl- and dolichyl- pyrophosphate lipid carriers used in this study. **c** SDS-PAGE analysis of purified FLAG-ScRft1, HsRft1-FLAG, HhAgl22 and HhAgl23 tagged with a cellulose binding domain (CBD). Proteins were detected by Coomassie blue-staining. Images are representative of two independent experiments. **d** Time course of translocation (i.e., M5GN2-PP-Phy hydrolysis by α1-2 mannosidase) in proteoliposomes using protein to phospholipid ratio (PPR) of ~2 mg/mmol. **e** Translocation rate as a function of different PPRs. Mannosidase-mediated hydrolysis M5GN2-PP-Phy in ScRft1-proteoliposomes was for 2 h at different PPRs. **f** Translocation of M5GN2-PP-Dol. Mannosidase-mediated hydrolysis by α1-2 mannosidase for 2 h in Rft1-proteoliposomes using PPR of ~2 mg/mmol. Data in **d**, **f** are mean ± SEM of three independent biological repeats and *P* values are from *t* test (unpaired and two-tailed).

with ScRft1, proteoliposomes reconstituted with HsRft1 showed almost 90% conversion of M5GN2 to M3GN2 after 2 h of incubation. In sum, our results demonstrated that ScRft1, HsRft1, as well as HhAgl23, catalyze translocation of M5GN2-PP-Dol or M5GN2-PP-Phy across the lipid bilayer of reconstituted liposomes.

## Rft1 protein is required for M5GN2 flipping

Previous biochemical assays of M5GN2-PP-Dol flipping in proteoliposomes reconstituted with ER membrane proteins from yeast or rat liver, in which Rft1 was reduced or removed, nevertheless displayed robust flippase activity[23,24,29]. These results raised doubts about a direct role for Rft1 in translocation.

To further investigate the discrepancy between the assay we describe here, in which purified Rft1 is required for M5GN2 translocation in vitro, and those assays previously described using yeast Rft1-depleted fractions, we took advantage of a *HRD3* high copy *rft1Δ* suppressor strain (*rft1Δ*+*HRD3*) identified during our *rft1Δ* suppressor screen (see Supplementary Fig. 6 for the detailed screening). Hrd3 is a component of the ubiquitylation machinery[38]. This strain, despite lacking *RFT1*, is able to grow though with a reduced rate compared to wild type (Fig. 3a). This strain also accumulates hypo-glycosylated CPY, indicating a defect in N-glycosylation (Fig. 3b). Moreover, the accumulation of M5GN2 intermediate in *rft1Δ*+*HRD3* at 12 h of incubation was analyzed by UPLC-MS, and its relative ratio (see "Methods" or Legend of Supplementary Fig. 7 for calculation) was compared to wild type or glucose-repressed *GALpr-RFT1* (Supplementary Fig. 7). Our results demonstrated that, unlike wild type cells, *rft1Δ*+*HRD3* accumulate the

M5GN2 intermediate (26.96%) though less than glucose-repressed *GALpr-RFT1* (49.87%) (Supplementary Fig. 7). Notably, accumulation of both hypo-glycosylated CPY and the M5GN2 intermediate suggested that this *rft1Δ*+*HRD3* strain is specifically defective in M5GN2-PP-Dol lumenal translocation (Fig. 3c). These results demonstrated that Hrd3 suppression of *rft1Δ* lethality involves a mechanism that bypasses the *rft1Δ*-dependent N-glycosylation defect. While the basis for Hrd3 suppression of *rft1Δ* is not yet understood, this strain (W303a-*rft1Δ*+*HRD3*) is viable despite lacking Rft1. This viability provided an experimental system to compare our assay reconstituted with purified Rft1 with one that is reconstituted with yeast membrane protein fractions lacking Rft1[24]. Membrane protein fractions were prepared from *rft1Δ*+*HRD3* or wild type strains and reconstituted into proteoliposomes to test M5GN2-PP-Dol flipping activity. The results of this experiment demonstrated that proteoliposomes reconstituted with a membrane fraction from the *rft1Δ*+*HRD3* yeast strain displayed no flippase activity, as evidenced by an undetectable decrease in the M5GN2 to M3GN2 ratio, with 50% of M5GN2 remaining after a 2 h incubation. This 1:1 M5GN2 to M3GN2 ratio was the same as in protein-free liposomes. In contrast, proteoliposomes reconstituted with membrane fractions from the parental wild type had less than 5% of M5GN2 remaining (Fig. 3d). These results provide further evidence to support a direct involvement of Rft1 in M5GN2-PP-Dol flipping.

## Rft1 is specific for M5GN2-PP-Dol translocation

The cytosolically localized mannosylation steps of the LLO biosynthetic pathway include the stepwise synthesis of M1-, M2-, M3-, M4- and

M5GN2-PP-Dol intermediates. Two key attributes ascribed to the M5GN2-PP-Dol ER flippase are its ER localization[39], and its preferential recognition of cytosolic M5GN2-PP-Dol compared to other LLO intermediates. As a test of ScRft1 substrate specificity, we assayed its flippase activity in liposomes containing M3GN2-PP-Phy as the substrate (see Supplementary Fig. 3 for preparation). To assay these M3GN2-PP-Phy containing liposomes, we used an α1-2,3 mannosidase to convert M3GN2 to M2GN2 as the readout to quantify translocation (Fig. 4a). UPLC-MS analysis of oligosaccharides prepared from these liposomes in the presence or absence of α1-2,3 mannosidase demonstrated that this treatment resulted in the predicted conversion of M3GN2 to M2GN2, with M2GN2 accounting for 50% of the oligosaccharide (Fig. 4a; Supplementary Fig. 8). Proteoliposomes reconstituted with purified ScRft1 revealed no decrease in the M3GN2 to M2GN2 ratio, with 50% of M3GN2 remaining after a 4 h incubation (Fig. 4b). In

contrast, in HhAgl23-containing proteoliposomes less than 15% of M3GN2 remained, validating HhAgl23 M3GN2-PP-Dol flipping activity (Fig. 4b). These results demonstrated that unlike HhAgl23, ScRft1 can translocate M5GN2-PP-Phy across the membrane much more efficiently than M3GN2-PP-Phy, suggesting its strong preference for the M5GN2 cytoplasmic LLO intermediate. These data provide additional evidence to support a direct role for ScRft1 as the M5GN2-PP-Dol flippase of LLO biosynthetic pathway.

## Discussion

The tremendous progress in methods for purification of recombinant glycosyltransferases and the unambiguous characterization of their glycan products was exploited in this study for development of a reconstituted assay for lipid-linked M5GN2 translocation in liposomes (Fig. 2a and Supplementary Fig. 4). Translocation was ATP-independent,

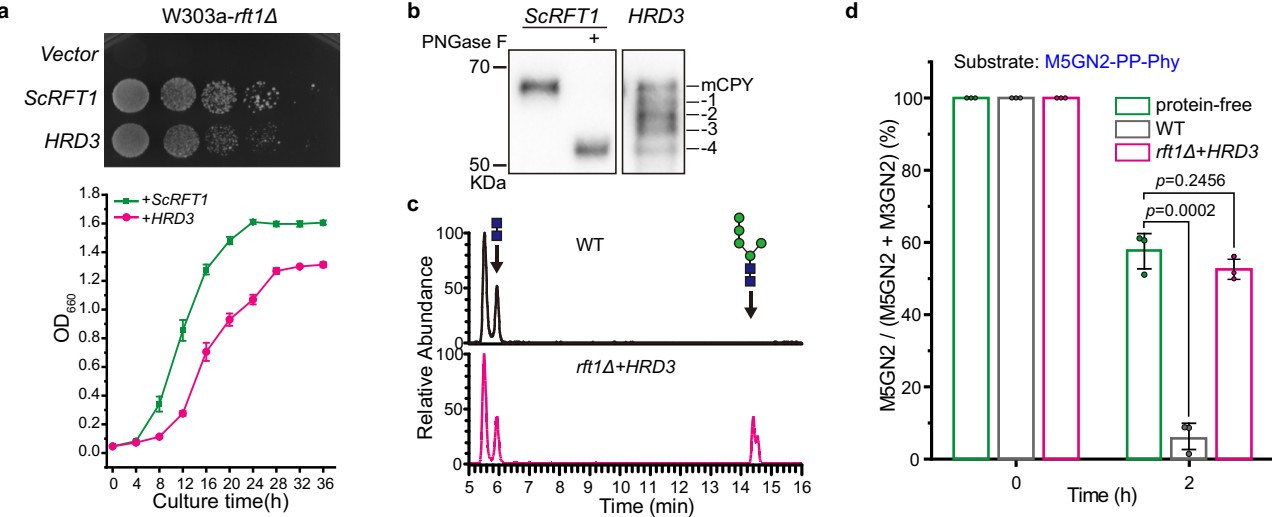

**Fig. 3 | Rft1 is required to translocate M5GNA2-PP-Phy across lipid bilayer.**
**a** Overexpression of *HRD3* rescues the growth defect of yeast *rft1Δ* cells. YEp351-GAPII plasmids, containing *ScRFT1*, *HRD3* or empty vector, were introduced in W303a-*rft1Δ* mutants containing pRS316-*ScRFT1*. Serial dilutions of each strain were plated on SD-Leu containing 5-FOA and incubated for 2 days at 30 °C (upper); growth curves of W303a- *rft1Δ+ScRFT1* and W303a- *rft1Δ+HRD3* cultured in YPAD at 30 °C (below). **b** Overexpression of *HRD3* fails to rescue the rft1Δ-dependent hypo glycosylation of CPY. The positions of fully glycosylated (mCPY) and hypoglycosylated CPY (lacking up to four oligosaccharides) are indicated in a western blot of

CPY from *rft1Δ* + *ScRFT1* or + *HRD3* strains. Images are representative of two independent experiments. **c** UPLC chromatograms of glycan chains released from the lipid-linked oligosaccharide intermediates from wild type or *rft1Δ+HRD3* cells. **d** M5GN2-PP-Phy hydrolysis by α1-2 mannosidase for 2 h in proteoliposomes reconstituted with membrane proteins extracted from WT (gray) and *rft1Δ+HRD3* cells (pink) using PPR of ~20 mg/mmol. Protein-free liposomes (green) were used as a negative control. Data in **a** (below) and **d** are mean ± SEM of three independent biological repeats and *P* values are from *t* test (unpaired and two-tailed).

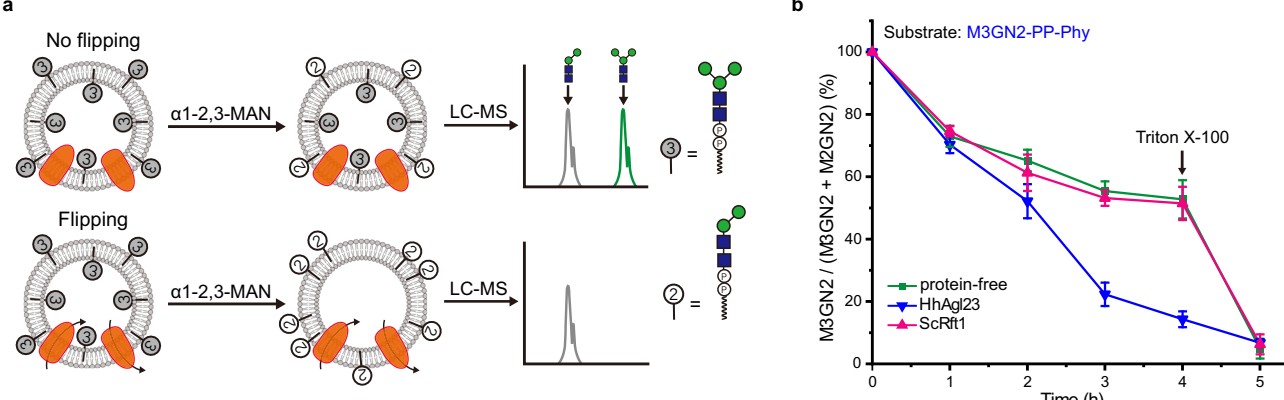

**Fig. 4 | Rft1 fails to translocate M3GNA2-PP-Phy across the lipid bilayer.**
**a** Schematic of α1-2,3 mannosidase-based translocation assay by UPLC-MS analysis. M3GN2-PP-Phy in the outer leaflet of liposomes is digested to M2GN2-PP-Phy by α1-2,3 mannosidase. In the absence of a flippase, there is a 1:1 distribution of M3GN2:M2GN2.

Flippase activity is measured as an increase in this ratio. **b** Translocation of M3GN2-PP-Phy. Mannosidase-mediated hydrolysis by α1-2,3 mannosidase in proteoliposomes using PPR of ~2 mg/mmol. Triton X-100 was added after 4 h incubation. Data in **b** are mean ± SEM of three independent biological repeats.

occurred with similar kinetics for both dolichol or phytanol lipid carriers, and importantly, was dependent on addition of purified Rft1 or archaeal Agl23 (Fig. 2d, f). Furthermore, proteoliposomes isolated from a *rft1Δ* yeast mutant whose viability is possible by the over-expression of *HRD3* lacked this activity. Together, these experiments provide clear evidence for a direct role of Rft1 in M5GN2-PP-Dol translocation of LLO pathway.

The direct versus indirect role of Rft1 in flipping ER LLO inter-mediates is a long-standing question. Previous biochemical studies assigned a M5GN2-PP-Dol flippase in ER membrane fractions prepared from yeast and *T. brucei* cells or from rat liver[23,26,30]. This M5-DLO flippase activity could be resolved from Rft1 by a variety of methods[23,24,40], including velocity sedimentation, ion exchange chro-matography and Con A-Sepharose. Moreover, it has been postulated that the Rft1-independent M5-DLO flippase activity is likely to be regulated by interaction with other molecules in a complex[41]. In con-trast, our results demonstrated that purified Rft1 is sufficient for translocation of both M5GN2-PP-Dol and M5GN2-PP-Phy (Fig. 2d, f). Another difference between the reported M5GN2-PP-Dol flipping activity and Rft1 is substrate specificity. A previous study of Rft1-independent flippase activity in proteoliposomes reconstituted with rat liver ER membrane proteins reported that the M3GN2-PP-Dol intermediate is translocated almost as well as M5GN2-PP-Dol[26]. In contrast, our in vitro assay indicated that purified ScRft1 showed a strong preference for M5GN2-PP-Phy (Fig. 4). The different properties of purified Rft1 activity that we describe here and those flippase activities described in previous studies[23,24,26,40,30] suggest that the putative flippase protein(s) assayed previously and Rft1 are molecu-larly distinct entities. In addition to its flippase activity (Figs. 1b and 3a), in vivo ER localization, and strong selectivity of M5GN2 over M3GN2 (Fig. 4), our results unequivocally support the idea that Rft1, from both yeast and human, is directly responsible for M5GN2-PP-Dol transloca-tion across the ER membrane during LLO biosynthesis.

Additional evidence that relates Rft1 flipping activity to its important function in yeast came from our genetic suppressor ana-lyses of *RFT1* orthologues and putative glycolipid flippases. We iden-tified two different archaea *H. hispanca* genes that suppressed the inviability of a yeast *rft1Δ* strain. The first, *HhRFT1*, is a structural *RFT1* orthologue, but the second, HhAgl23, bears no resemblance to Rft1 (Supplementary Fig. 2). HhAgl23 complements both *rft1Δ* inviabilty, and its N-glycosylation defect (Fig. 1). Moreover, purified HhAgl23 protein displays M5GN2-PP-Dol flipping activity (Fig. 2d). The simplest explanation for these observations is that complementation of *rft1Δ* lethality by HhAgl23 is directly related to its M5GN2-PP-Dol flipping activity. This implies that the lethality in *rft1Δ* strain is caused by loss of M5GN2-PP-Dol flipping, further demonstrating that Rft1 directly cata-lyzes this important function in vivo.

The idea that Rft1 is an M5GN2-PP-Dol flippase in yeast can be reconciled with the unexpected phenotype of the *rft1Δ* strain whose growth defect is partially suppressed by high copy expression of *HRD3* (Fig. 3a). If M5GN2-PP-Dol translocation into the lumen is required for viability, then *HRD3* suppression of *rft1Δ* should be accompanied by an increase in M5GN2-PP-Dol flipping. As predicted, analysis of CPY gly-cosylation in this viable *rft1Δ+HRD3* strain revealed that despite the loss of *RFT1*, a small amount of fully glycosylated CPY (Fig. 3b) could be detected. This indicates that some M5GN2-PP-Dol intermediate was translocated into the ER lumen and that this level of M5GN2-PP-Dol ER lumenal translocation is sufficient for bypassing lethality. Presumably, this residual translocation of M5GN2-PP-Dol involves some type of compensatory mechanism triggered by the overexpression of *HRD3* in these *rft1Δ* cells. We speculate this may be due to an alteration of the integrity, structure, or protein abundance of the ER membrane that in turn affects its permeability or the activity of other flippases or scramblases such that a small amount of M5GN2-PP-Dol (as well as other mannosylated intermediates) is flipped. The idea that an

alteration of ER membrane properties is involved in the M5-flipping bypass mechanism in vivo is supported by our observation that in vitro, liposomes reconstituted with proteins extracted from this *rft1Δ+HRD3* strain failed to display any flippase activity. These results also refute the idea that putative Rft1-independent M5GN2-PP-Dol flippases exist that can catalyze this function in yeast. The molecular mechanism of such a *HRD3*-dependent compensatory mechanism is of interest whose understanding requires further investigation.

The discovery that *H. hispanca* encodes at least two partially redundant flippases that can complement yeast *rft1Δ* raises the pos-sibility that during the evolution of N-linked glycosylation, some eukaryotes may similarly retain redundant flippases in their genomes. If correct, this idea could explain some of the discrepancies in the literature pertaining to Rft1 and its role as the M5GN2-PP-Dol flippase. In addition to early experiments describing the role of Rft1 in ER N-glycosylation of yeast and mammalian cells, several studies exam-ined LLO synthesis in a *Trypanosoma brucei rft1Δ* mutant. This is because *T. brucei* is not dependent on *RFT1* for viability or normal N-glycosylation[39,30]. Indeed, the viability of the *T. brucei rft1Δ* mutant has been interpreted as evidence that Rft1 indirectly affects flippase activity. While speculative, our identification of archaeal Agl23 as a yeast *rft1Δ* suppressor raises the possibility that Trypanosomes, which represent early diverging eukaryotes, may also contain another M5GN2-PP-Dol flippase that is partially redundant with Rft1. This could explain why *T. brucei rft1Δ* is viable and synthesizes normal levels of mature LLOs. There are also example of several other protozoans, including *Perkinsus marinus, Gregarina niphandrodes, Toxoplasma gondii* and *Plasmodium*, that possess orthologues of most *ALG* (asparagine-linked glycosylation) glycosyltransferases yet lack *RFT1*[3]. These species lacking Rft1 may have co-opted one or more proteins to assist in M5GN2-PP-Dol lumenal ER translocation. The discovery that Rft1 and archaeal Agl23 are functionally similar as M5GN2- PP-Dol flippases will allow a test of these ideas and open new avenues for future studies of M5GN2-PP-Dol translocation across the ER mem-brane. We note during final preparation of our manuscript, a preprint on Rft1 was published[42].

## Methods
### Plasmids, strains, and media
Standard molecular biology techniques were used for all plasmid constructions. The correct sequence of all PCR-amplified products was verified by DNA sequencing and the sequences of primers used in this study are available upon request.

To construct pRS316-RFT1, the ORF of yeast *RFT1* (*ScRFT1*) was cloned into a *CEN6/URA3* vector (pRS316)[43]. Candidate suppressor genes, including human *RFT1* (*HsRFT1*, NC_000003.12), *Trypanosoma brucei RFT1* (*TbRFT1*)[30], *Haloarcula hispanica RFT1* (*HhRFT1*, WP_023842997.1), *Haloferax volcanii RFT1* (*HvRFT1*, WP_004043121.1), human *CLPTM1L* (*HsCLPTM1L*)[31], *Thermosipho africanus MURJ* (*TaMURJ*)[44], *Haloarcula hispanica AGL23* (*HsAGL23*) and *AGL22* (*HsAGL22*)[33] were PCR-amplified and cloned in a *2μ/LUE2* yeast shuttle vector (YEp351-GAPII)[45] for expression in yeast. To construct YEp351-GAPII-FLAG-RFT1, a FLAG tag was fused at the N-terminus of *RFT1* in YEp351-GAPII-RFT1. The construction of pWL-CBD-AGL22 and pWL-CBD-AGL23 is described[33].

*S. cerevisiae* W303a (*MATa ade2-1 ura3-1 his3-11 trp1-1 leu2-3,112 can1-100*) and BY4741 (*MATa his3Δ1 leu2Δ0 met15Δ0 ura3Δ0*) were used as the parental yeast strains for all experiments. The *rft1Δ* strain was generated by introducing plasmid borne *RFT1* (pRS316-RFT1) into W303a and then replacing the chromosomal *RFT1* with a PCR fragment amplified from the kanMX4-marked pFA6a-kanMX6 deletion cassette[46]. The W303a-GALpr-RFT1 strain contains a replacement of the *RFT1* chromosomal promoter with the *GAL1-10* promoter from the pFA6a-TRP1-GAL1 plasmid[46]. Archaea strain *Haloarcula hispanica* ATCC 33,960 (*Hh*) was used as the parental strain for *H. hispanica*

strain constructions. Strain *Hh-agl23Δ* was created using homologous recombination (pop-in/pop-out) method, as described[33,47]. Standard yeast medium and genetic techniques were used[48]. *H. hispanica* was cultured at 42 °C in 23% salt AS-168 medium containing 1% agar if needed, as described[49].

Plasmids and strains, including their description are listed in Supplementary Tables 1 and 2.

## Preparation of protein extracts and western blotting
Protein extraction from yeast was conducted as follows. Cells were lysed as described[9,50], and the lysates were centrifuged at 4000 × *g* for 15 min to remove unlysed cells and cell wall debris. The supernatant was collected as "total protein". Protein concentration was determined by Nano-Drop (Thermo-scientific, MA, USA). For western blotting, 25 μg protein was subjected to 10% SDS-PAGE and transferred to PVDF membranes (Bio-rad, Shanghai, China). Mouse anti-FLAG (Transgen, Beijing, China) or CPY (Abcam, Shanghai, China) antibodies were used at 1:2000 dilution. Goat anti-mouse IgG-HRP (Transgen, Beijing, China) was used at 1:4000 dilution. Signals were visualized with Clarity Western ECL Substrate (Bio-rad, Shanghai, China) and images were obtained by using Image 5200Multi (Tanon).

## Expression and purification of recombinant proteins in *E. coli*
The *E. coli* Rosetta (DE3) strain (Thermo Scientific, MA, USA, Catalog Number: 70-954-4) was used as the host for expression of recombinant proteins: (1) The His-tagged α1-2 mannosidase (BT3990) was expressed, extracted, and purified with a HisTrap HP affinity column (GE Healthcare Life Sciences) as described previously[7]. The purified α1-2 mannosidase was dialyzed against buffer [10 mM HEPES-NaOH, pH 7.4, 100 mM NaCl], followed by concentration using Amicon Ultra 10 K NMWL filtration units (Millipore, MA, USA). Protein concentration was determined with the BCA assay kit (Sangon Biotech, Shanghai, China); (2) For enzymatic synthesis of LLO intermediates, yeast Alg1ΔTM, Trx-Alg2 and Alg11ΔTM genes were co-expressed. Membrane fractions containing recombinant Alg1ΔTM, Trx-Alg2, and Alg11ΔTM were prepared as described previously[7]. These membranes were homogenized in [50 mM Tris/HCl (pH 7.5), 30% glycerol] and stored at −20 °C. Enzyme activity in membranes remained active for at least 3 months.

## Enzymatic assembly and purification of M3GN2-PP-Phy, M5GN2-PP-Phy and M5GN2-PP-Dol
Chemical synthesis of GN2-PP-Phy and GN2-PP-Dol was performed as reported[51,52]. All enzyme assays were performed in the following buffer: [14 mM MES/NaOH (pH 7.0), 4 mM potassium citrate, 10 mM MgCl₂, 0.05% NP-40, 50 μM GN2-PP-Phy/GN2-PP-Dol, 1 M sucrose; 2 mM GDP-Man] in a total volume of 100 μL. For assembly of the M5GN2-PP-Phy and M5GN2-PP-Dol, reactions performed with membrane fractions from *E. coli* that co-expressed Alg1ΔTM/Trx-Alg2/Alg11ΔTM (20 μg/mL) were incubated at 30 °C for 24 h. Similarly, the M3N2-PP-Phy was prepared by using the membrane fraction co-expressed with Alg1ΔTM/Trx-Alg2. M3N2-PP-Phy, M5GN2-PP-Phy and M5GN2-PP-Dol produced in these reactions were purified as follows: 1 mL chloroform/methanol (2:1) was added to the freeze-dried reaction mixture that was solubilized by ultrasonic treatment for 10 min with occasional vortexing. This was centrifuged at 3000 × *g* for 10 min and the supernatant discarded. The precipitate was resuspended in 300 μL methanol and blown dry with nitrogen. The precipitate was resuspended by 1 mL H₂O, treated again with ultrasound and vortexing for 10 min, centrifuged 3000 × *g* for 10 min and the supernatant discarded. The resulting precipitate was resuspended with 300 μL methanol and blown dry with nitrogen. This precipitate was resuspended by 1 mL CMW (10:10:3), solubilized by ultrasound and vortex for 10 min, spun at 3000 × *g* for 10 min. The resulting supernatant containing lipid-linked M5GN2 was collected, freeze-dried, and stored at −80 °C.

## Extraction of cellular lipid-linked oligosaccharides (LLOs)
Lipid-linked oligosaccharides were extracted from various yeast strains as reported[53] with some modification. Briefly, -10⁹ cells were harvested and suspended in 4 mL chloroform/methanol 3:2 (CM). After adding 0.5 ml of acid-washed glass beads, cells were vortexed for 1 min followed by washing twice with 4 mL CM. The collected cell pellet was further washed three times [chloroform: methanol: water (CMW), 3:48:47, v/v/v containing 4 mM MgCl₂] and once (CMW, 3:48:47, v/v/v). Lipid-linked oligosaccharides were then extracted with 3×4 mL CMW (10:10:3, v/v/v), and the extraction solutions were dried under nitrogen at 37 °C. For hydrolysis, the dried lipid-linked oligosaccharides were suspended in 35 μL I-propanol, followed by the addition of 1 mL of 20 mM HCl and incubated at 100 °C for 60 min. For desalting, the water-soluble glycan-containing fraction was applied for a solid-phase extraction using 1 mL of Supelclean ENVI-Carb slurry (Sigma) equilibrated with 2% acetonitrile. After washing the column with 2% acetonitrile (10 mL), oligosaccharides were eluted with 25% acetonitrile (3.0 mL) and lyophilized. These LLOs were analyzed by UPLC-MS.

Of note, for preparing LLOs from the *GALpr-RFT1* strain (Figs. 1d and 3c), yeast was grown at 30 °C in YPAD (Glc) or YPAG (Gal) medium to a density of 1-3 × 10⁷/ml.

## UPLC-MS analysis of oligosaccharides
Dried samples were dissolved in 80% acetonitrile and applied to UPLC–MS analysis using the Dionex Ultimate 3000 UPLC (Thermo Scientific, MA, USA). Glycans were injected into an Acquity UPLC BEH Amide column (1.7 μm, 2.1 × 100 mm, Waters, MA, USA) and eluted with an acetonitrile gradient with a flow rate 0.2 mL/min. The gradient program was set as follows; 0–2 min, isocratic 80% acetonitrile; 2–15 min, 80–50% acetonitrile; 15–18 min, isocratic 50% acetonitrile. The ESI-MS of eluate was measured on a TSQ Quantum Ultra (Thermo Scientific, MA, USA) in the mass range of 400–2000 (m/z, positive mode). The relative content oligosaccharide was quantified by calculating the peak intensity in LC-ESI-MS using Xcalibur (Version 2.0, Thermo Scientific, USA).

## Purification of FLAG-ScRft1, HsRft1-FLAG, CBD-HhAgl22 and CBD-HhAgl23
W303a cells harboring YEp351-GAPII-*FLAG-ScRFT1* and YEp351-GAPII-*HsRFT1-FLAG* were grown in SD-Leu to OD₆₆₀ ≈ 1.0. Yeast harvested from 200 mL were suspended in 5 mL lysis buffer [25 mM HEPES pH 7.4, 150 mM NaCl, 1% DDM (n-Dodecyl-β-D-maltoside, Macklin Bio. Tech., Shanghai, China) and 1× protease inhibitor cocktail (Roche/ Mannheim, Germany)], and lysed by vortexing with glass beads on ice. The homogenate was centrifuged at 20,000 × *g* to remove the pellet fraction. 0.1 mL anti-FLAG resin (F3290, Sigma Aldrich) was added to the resulting supernatant and incubated on a nutator for 4 h at 4 °C. After three 1 mL washes (25 mM HEPES, 150 mM NaCl, 0.1% DDM, pH 7.4), the anti-FLAG resin was collected, resuspended in 0.2 mL TBS buffer containing 500 μg/mL FLAG peptide (P9801, Beyotime, Shanghai, China) and rotated for 2 h at 4 °C. The supernatant was collected and further fractionated by gel filtration (Superose 12 HR 10/30, Pharmacia) equilibrated with elution buffer. The peak Rft1-containing fractions were pooled, concentrated and then stored at −80 °C.

Cellulose binding domain (CBD)-fused proteins from archaea *H. hispanica* were isolated by the Capture CBD method as described previously with slight modification[54]. Briefly, 2 mL aliquot of *H. hispanica* cells (OD₆₀₀ ≈ 1.0) were harvested at 4000 × *g* in a microfuge for 10 min, and the pelleted cells were resuspended in 2 mL solubilization buffer (1% Triton X-100, 1.8 M NaCl, 50 mM Tris-HCl, pH 7.2) for 2 h. added 50 μL of a 10% (w/v) solution of cellulose was added and incubated for 60 min with rotation at 25 °C. The mixture was centrifuged (20,000 × *g* for 10 min) and the cellulose pellet washed twice with washed twice with [2 M NaCl, 50 mM Tris-HCl, pH 7.2]. CBD-fused

proteins were eluted by 1 mL Elution buffer (50 mM Tris-HCl, pH 8.0) and stored at −80 °C.

## Proteoliposome reconstitution

Liposomes and proteoliposomes were prepared according a previously reported one-pot method using a mixture of lipids including egg PC and brain PS at a molar ratio of 9:1[23,35,36]. Briefly, 3.6 mg of PC (Sinopharm Chemical Reagent Co.) and 0.4 mg of PS (Guangzhou Weihua Biotech. Co) were mixed in a glass tube, dried under nitrogen for 30 min to remove chloroform, and dried by desiccation for 1 h. The dried lipids were solubilized in 900 μL of buffer A (10 mM HEPES-NaOH, pH 7.4, 100 mM NaCl, 1% (w/v) TX-100) contained 5 nmol M3GN2-PP-Phy, M5GN2-PP-Phy or M5GN2-PP-Dol, vortexed at moderate speed for 20 min until the solution became completely transparent. To make proteoliposomes, 100 μL of purified proteins including Rft1, Agl23 and Agl22 or membrane fraction extracted from yeast cells was mixed with prepared protein-free liposomes in 900 μL of buffer A to make total volume of 1000 μL. The reaction mixture was incubated for 1 h at 4 °C with end over end rotation. To eliminate detergents, 100 mg of wet hydrophobic SM-2 Bio-Beads (Bio-Rad) was added and rotated for 2 h at room temperature. Moreover, the supernatant of reaction mixture was collected and incubated with another 200 mg of Bio-beads for 16 h at 4 °C followed by a third round of incubation with 200 mg Bio-beads for 6 h at 4 °C. After removing the beads, the stabilized liposomes in supernatant were used immediately for translocation assays. The recovery of phospholipid and protein were determined as described[55]. The phospholipid recovery rate of protein-free liposomes is 61.50 ± 3.09%, while ScRft1 proteoliposomes revealed 59.54 ± 1.73% phospholipid recovery and 39.54 ± 2.6% protein recovery. Similar analyses for HhAgl23, HhAgl22 and HsRft1 proteoliposomes, the protein recovery was 55.10 ± 2.5%, 60.04 ± 4.41%, and 42.16 ± 5.44%, and the corresponding phospholipid recovery rate were 64.19 ± 0.98%, 61.30 ± 1.17% and 60.91 ± 0.50%. The protein/phospholipid ratio (PPR; mg of protein/mmol of phospholipid) of the vesicles was determined as previously described[55].

## Translocation activity assay

To assay the in vitro translocation of LLO, 100 μL of liposomes or proteoliposomes containing M3GN2-PP-Phy, M5GN2-PP-Phy or M5GN2-PP-Dol was mixed with 10 μL of reaction buffer (10 mM HEPES-NaOH, pH 7.4, 100 mM NaCl, 1 mM CaCl₂) containing 50 μg of mannosidase. The reaction mixture was incubated at 30 °C. At the indicated time, reactions were stopped by heating at 95 °C for 5 min. An equal volume of 40 mM HCl was added and the reaction further incubated at 100 °C for 60 min to obtain water-soluble oligosaccharides[53]. The water-soluble glycan-containing fraction was desalted by solid-phase extraction using 1 mL SupelcleanENVI-Carb Slurry (Sigma-Aldrich, MO, USA) and lyophilized. Dried samples were dissolved in 80% acetonitrile and analyzed by UPLC-MS[50].

## Extraction of membrane proteins

Yeast cells were grown in YPAD to $OD_{660} ≈ 1.0$. Yeast harvested from 200 mL were suspended in 5 mL lysis buffer [25 mM HEPES pH 7.4, 150 mM NaCl and 1 × protease inhibitor cocktail (Roche/Mannheim, Germany)], and lysed by vortexing with glass beads on ice. The homogenate was centrifuged at 1000 × g, 4 °C for 5 min to remove unbroken cells and cell fragments, 3000 × g, 4 °C for 5 min to further remove cell fragments and nuclear components, and 20,000 × g, 4 °C for 90 min to collect the membrane component. After washing the precipitate with 5 mL lysis buffer for three times, this membrane fraction was resuspended into 5 mL lysis buffer with 1% DDM and directly used to construct liposomes. The concentration of membrane proteins was detected with BCA kit. The PPR of proteoliposomes reconstituted with these membrane proteins was ~20 mg/mmol.

## PiggyBac (PB) transposon-based mutagenesis and screening

To construct BY4741-tet-off-RFT1-PB, the endogenous RFT1 promoter was replaced with a doxycycline (Dox) "tet-off" promoter by homologous recombination. In this strain, the cabbage looper moth TTAA-targeting transposon PB, marked with HIS3 (3'PB-P_TEF-HIS3-5'PB)[56] was inserted into a TTAA sequence in ADE2, whose red/white color phenotype was used as a marker for transposon excision (see Supplementary Fig. 6 for detail). The galactose-inducible transposase on a URA3-marked plasmid, pRS316-GALpr-PBase[56] was transformed BY4741-tet-off-RFT1-PB and cultured in liquid SD-Ura medium for 1 day. Yeast cells were suspended into 25 mL SD-Ura medium supplemented with galactose at a final concentration of $OD_{660} = 0.025$ and cultured for an additional 27 h to allow the expression of the PBase transposase. Subsequently, 1 mL of yeast cells ($OD_{660} = 0.8$) was transferred to 25 mL SD-Ade-His medium and cultured at 30 °C for 24 h. Then, 1 mL ($OD_{660} = 0.5$) of cells were again diluted and cultured in SD-Ade-His medium for an additional 36 h. To select mutants that could grow in the absence of RFT1, cells were plated on SD-Ade-His + Dox plates (about 5-6 × 10³ cells/plate × 10). Fast growing colonies were picked for subsequent validation and PB insertion site sequencing as described[51,57].

Quantitative RT-PCR methods used for quantification of HRD3 mRNA were performed as described[56]. HRD3 mRNA expression levels were normalized to those of the constitutively expressed TDH1.

## Statistical analysis

All statistical analyses were performed using OriginPro 2021. Error bars and sample size for each experiment are indicated in figure legends. Comparisons between two individual groups were analyzed by test, and $P < 0.05$ was considered statistically significant.

## Reporting summary

Further information on research design is available in the Nature Portfolio Reporting Summary linked to this article.

# Data availability

The data that support this study are available from the corresponding authors upon request. The data supporting the findings of this study are available within the Article and its Supplementary Information. A Source Data file is available with this paper. Source data are provided with this paper.

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

## Acknowledgements

We are grateful to members of the Gao laboratory for reagents, comments and other contributions to this project. This work was supported by grants-in-aid from the National Natural Science Foundation of China (31971216; 32271342), National First-Class Discipline Program of Light Industry Technology and Engineering (LITE2018-015), Shandong Provincial Major Scientific and Technological Innovation Project (2019JZZY011006) and Special Fund for Zaozhuang Excellence Agglomeration Project to X-D Gao.

## Author contributions

Conceptualization, X.-D.G., D.N, and C.J.; methodology, S.C., C.-X.P., S.X., H.L., and Y.-S.L.; validation, S.C., Y.W., and Y.-S.L; formal analysis, S.C., C.-X.P, S.X., and H.L.; investigation, S.C., Y.W., and X.-D.G.; writing—original draft preparation, S.C., C.-X.P., and X.-D.G.; writing—review and editing, N.D., C.J., and X.-D.G.; supervision, X.-D.G., N.D., and C.J.; funding acquisition, X.-D.G., N.D., and C.J. All authors have read and agreed to the published version of the manuscript.

## Competing interests

The authors declare no competing interests.
