## [Peer Review File · Nature Communications]

Rft1 directly catalyzes lipid-linked oligosaccharide translocation across the ER membraneReviewer #1 (Remarks to the Author):

This is a provocative, potentially important paper concerning the function of Rft1, an essential membrane protein in yeast and humans. A yeast genetics study in 2002 suggested that Rft1 flips the glycolipid Man5GlcNAc2-PP-dolichol across the ER membrane, a critical step in the assembly of the oligosaccharide precursor necessary for protein N-glycosylation. Subsequent biochemical studies indicated that this was not the case. Of relevance to the current paper, some of these studies recapitulated Man5GlcNAc2-PP-dolichol flipping in vesicles reconstituted with ER membrane proteins from yeast or rat liver and showed - using multiple techniques - that the flipping activity could be convincingly resolved from Rft1. These studies concluded that Rft1 is not necessary for flipping Man5GlcNAc2-PP-dolichol. These studies also showed that flipping is highly specific for Man5GlcNAc2-PP-dolichol, and very fast, occurring on the order of 1 minute.

The present paper claims that Rft1 is indeed the Man5GlcNAc2-PP-dolichol flippase by (i) identifying an archaeal protein (Agl23), a putative hexose-P-dolichol flippase, that can rescue Rft1 deficiency in cells, (ii) demonstrating the flippase activity of both Rft1 and Agl23 after reconstitution into vesicles, and (iii) using an enigmatic strain lacking Rft1 but over-expressing Hrd3, which retains viability but does not appear to have Man5GlcNAc2-PP-dolichol flippase activity when tested via biochemical reconstitution. There are numerous technical and conceptual problems with the paper.

Detailed comments are provided below. The main issues concern the authenticity of the purified proteins used for reconstitution and assay, lack of specificity controls to validate the assay, incomplete description of methods, and errors in describing previous work. The latter point derails the authors attempt to reconcile their results with previous reports.

1. Figure 1d and 3c. The oligosaccharide profiles are strange. Only GlcNAc2 is seen in WT (and rescued) cells, whereas a more complete complement of LLOs would be expected, especially Man5GlcNAc2, Man9GlcNAc2, and Glc3Man9GlcNAc2. Indeed, it is surprising to see GlcNAc2 at all, as GlcNAc2-PP-dol would be expected to be removed in the less polar solvent (CM 3:2) used for initial extractions. These points need to be addressed. Also: relative abundance is not an adequate readout here as it should be demonstrated more exactly how much Man5GlcNAc2-PP-dol accumulation in the glucose-grown GAL-Rft1 cells compared with WT.

2. Figure 2c. It is the experience of this reviewer that Rft1 runs anomalously fast on SDS-PAGE gels, below the 50 kDa marker. The bands in the figure corresponding to FLAG-tagged ScRft1 and HsRft1 are in the 60-70 kDa range, at the size predicted from sequence, but not corresponding to how the protein behaves on SDS-PAGE. There is something critically wrong here and this point must be resolved. It is also very surprising to learn that the authors lyophilized their protein preps (lines 380 and 388) - please clarify.

3. Figure 2d. The kinetics of mannosidase treatment are very slow, with a half-time of 30 min judging from the liposome data. For vesicles reconstituted with ScRft1 and Agl23, the half-time of conversion of M5 to M3 appears to be about 1 hr. If the kinetics of mannosidase treatment and flipping are indeed resolved as suggested by these numbers (flipping being slower than mannosidase probing), then flipping in this assay is much slower than previously reported (line 54). The authors should address this issue.

4. Figure 2e. Reconstitution of a purified protein using the Triton/BioBead protocol of Rigaud (ref 36) results in unilamellar vesicles in the 150-200 nm size range. Protein occupancy during reconstitution is dictated by Poisson statistics. For a protein of the size of Rft1, Poisson analysis indicates that at a PPR of ~0.2 mg/mmol, 63% of the vesicles should have 1 or more flippases. Inspection of the data in Figure 2e indicates that the authors need to go to 15x higher PPR (3 mg/mmol) to achieve this level of occupancy. This is an important issue that the authors need to address. Also: how was the PPR determined (line 406 cites REF without providing a reference) and what were the protein and lipid

recoveries after reconstitution?

5. Figure 2d-f. There are no specificity tests to know if the activity that the authors describe matches their recapitulation of published reports that flipping has 'exquisite substrate specificity' (line 54-55). This would be an important validation of the assay being used and would also address the issue of whether Agl23 - a putative Hex-P-Dol transporter which would not have been expected to recognize a pyrophosphate-containing lipid - has similar specificity in this assay when compared with ScRft1 or HsRft1. Following the approach described in ref 26, the authors could therefore test other oligosaccharide-PP-dol species that are susceptible to the mannosidase used in their assay. They could also test structurally unrelated lipids such as glycerophospholipids using standard assays (for example, described by Wang et al. for CLPTM1L).

6. Extended Data Figure 4. The result that increased expression of HRD3 can rescue the loss of Rft1 via the Tet-off system is intriguing. The data documenting this are not clearly explained. Extended Data Figure 4b is not at all clear - the explanation needs to be written out properly and a better figure needs to be presented. No sequence data are provided to support this result. It is surprising that despite switching to a TEF promoter, HRD message levels are only increased 1.5-fold. A protein blot should be shown to see exactly the increase in level of HRD3 protein that rescues the Tet-off RFT1 strain when grown on Dox. Can HRD overexpression from a plasmid rescue glucose-grown PGAL-RFT1 cells?

7. Figure 3b. The cells being used are presumably Tet-off RFT1 grown on Dox, and not *rft1*-delta as used in Figure 1 - the authors should make this clear. HRD over-expressing cells show hypoglycosylated CPY - this is true, but it is also clear that CPY is indeed glycosylated with bands corresponding to 1, 2, 3 and 4 N-glycans. The authors claim (based on panel d) that this strain has no Man5GlcNAc2-PP-dol flippase activity and yet CPY receives N-glycans! This is an important paradox that the authors need to address.

8. Figure 3d. This panel describes an assay done on vesicles reconstituted with a membrane protein extract. How was this extract prepared, characterized and quantified? What PPR was used in the reconstitution?

9. Figure 3. As the authors are already preparing protein extracts and reconstituting them, it would help the controversy in this field if they could prepare an extract from FLAG-ScRft1-expressing yeast, and test activity after passing the extract over anti-FLAG resin to remove the tagged protein specifically. This would match the type of experiment previously reported in the literature, for example ref 24.

10. The authors make numerous errors in describing the previous biochemical data on Rft1.

(i) Many of the previous experiments were done on extracts from which Rft1 was eliminated (not simply depleted) - these extracts had full activity. For example, Ref 24 tested an extract from yeast cells expressing Rft1-Protein A in place of endogenous Rft1. Quantitative removal of this protein on IgG beads did not affect the Man5GlcNAc2-PP-dol flippase activity of the extract. Similarly, ref 23 fractionated an extract on blue dye resin and showed that the activity eluted at a medium salt concentration distinct from the Rft1 elution profile. Thus, the authors' repeated proposal (for example, lines 60, 161, 164-165, 203 and perhaps other places as well) that previous work was affected by incomplete removal of Rft1 is not correct.

(ii) Of note, some of the previous biochemical work was done on membranes or extracts prepared from glucose-grown cells expressing Rft1 under control of a GAL promoter. Even here, readouts such as the protein-dependence plots shown in Ref 24, are sensitive to the amount of remaining Rft1 - the shape of the plots would have changed in a predictable way, yet no effect was observed. The authors need to consider this point carefully as it does not offer the 'out' that they propose to resolve the discrepancy between their data and previous reports.

(iii) It is interesting to note that Rft1 is itself not a glycoprotein. Yet a previous report (PMID

20534553) shows that the protein(s) responsible for Man5GlcNAc2-PP-dol flippase activity bind to Con A Sepharose. This shows a completely different way of resolving Rft1 from Man5GlcNAc2-PP-dol flippase activity, unrelated to genetic depletion or immunodepletion protocols noted in points (i) and (ii).

(iv) Also of note, a recent paper (PMID 33446867) - which the authors should cite - used an orthogonal approach to identify Man5GlcNAc2-PP-dol flippase candidates. Several proteins were identified, but not Rft1 whose chromatography profile did not match that of the activity.

(v) Lines 204-205 state that previous experiments were done on liposomes reconstituted with proteins from an RFT1-null trypanosome strain. This is not correct. Similarly, lines 211-212 are incorrect.

Other points

1. The first section of the results (line 72) initially gives the impression that a suppressor screen was carried out, whereas the authors simply picked a selection of candidates to test. Group I candidates are Rft1 orthologs, and all of these can functionally substitute for yeast Rft1 as expected. Although the authors did come up with Agl23 as a suppressor, the choice of Group II candidates is poorly justified as the Man5GlcNAc2-PP-dolichol flippase activity is known to be highly specific (line 55 and cited references). Thus, the flippase is unlikely to transport substrates with a monophosphate linkage (i.e., substrates of CLPTM1L and Agl23), or substrates with an unsaturated alpha-isoprene (substrate of MurJ) and so it is not clear why these proteins were chosen! CLPTM1L and MurJ do not complement the rft1-delta strain - but are these proteins expressed in yeast? if expressed, are they in the right compartment?

2. Figure 2. The data appear to be technical replicates - three measurements done on the same batch of liposomes. This is insufficient. Unique protein preps, and unique reconstitutions should be done to reproduce the data.

3. The description of proteoliposome reconstitution (line 390) is inadequate and makes little sense as written. For example, lipids are dried, dissolved in detergent and then extruded (lines 395-397)? This would not result in liposomes! Were the proteoliposomes also extruded?

4. The activity assay (line 408) needs to be written out more clearly. The samples are prepared in Hepes/NaCl, but then diluted into MES/CaCl₂ -for assay - this is strange. Why was the buffer changed? At the end of the assay, it appears that the samples are heated at 95C, then HCl is added, and the samples are further heated. Is this treatment sufficient to capture all glycans? Why not extract the lipids first (as in Fig 1d) and then treat with HCl to obtain glycans? Line 414 - oligosaccharides were purified - how?

5. Figure 2d. Although Extended Data Fig 3 shows exemplary chromatograms related to mannosidase treatment of liposomes (what are the incubation times going from step II to II, and III to IV? this information is not provided in the legend), a full data set of UPLC chromatograms should be provided to accompany the traces in Fig. 2d. What fraction of the Man5GNPPlipid committed to the flippase experiment is recovered in the UPLC chromatograms as Man3GN2 and Man5GN2?

6. Figure 2d,e,f: The y-axis label needs to be changed. "The ratio of M5-GN2 (%)" does not make sense. Presumably the authors mean $100 \times M5 / (M3 + M5)$

7. A strain list needs to be provided and the authors need to be clear about what strains are used in each of the experiments. For example, note the ambiguity in point 7 (main comments).

8. The last line of the abstract is a little over the top! Even if the authors are correct in their conclusions, the flippases for Man-P-dol and Glc-P-dol, necessary for the conversion of Man5GlcNAc2-PP-dolichol to the canonical oligosaccharide donor Glc3Man9GlcNAc2-PP-dolichol, remain to be identified.

9. These points need to be corrected/addressed.

Line 43: not all eukaryotes use a 14-sugar oligosaccharide.

Line 46: sugars are added to Dol-P, not Dol-PP (-PP- is formed when GlcNAc-P is added to Dol-P as shown in Fig 1A).

Line 55: replace 'Such a' with 'A putative'.

Line 59: proteoliposomes reconstituted from detergent-solubilized yeast Er membrane proteins.

Line 61: the eukaryote/protist Plasmodium, has an N-glycosylation pathway but lacks Rft1 - this point should be added.

Line 85: GlcN-PI AND phospholipids

Line 89: it would be useful to state here that Agl22 is a glycosyltransferase.

Lines 166-167: 'Rft1 is strictly required ...' - this statement is not justified based on Fig 2, where Agl23 appears to perform the same function as Rft1

Line 197: this is not a screen, simply a test of candidates - the sentence should be modified.

Line 224: how many hours in glucose?

Line 238: is this a Coomassie-stained gel? Please specify.

Line 365: what is 'oligosaccharide transfer rate'?

Reviewer #2 (Remarks to the Author):

The study by Chen and colleagues provides a clear answer to a long standing and important biochemical question: The possible role of Rft1 in flipping the lipid-linked oligosaccharide (LLO) intermediate from the cytoplasmic to the luminal side of the ER membrane during synthesis of the donor substrate for N-linked glycosylation of proteins in eukaryotes.

The authors elegantly combine genetic screens with a novel biochemical flipping assay to demonstrate unequivocally that Rft1 is the flippase – as was reported by Helenius and Aebi based on genetic studies over 20 years ago and subsequently brought into question by biochemical studies. Using genetic screens they identify archaeal proteins that suppress lethality of a $\Delta rft1$ mutation in yeast. They proceed to develop a flippase assay that uses glycoside hydrolyse activity to modify LLOs that are exposed on the external surface of reconstituted liposomes. Because the LLOs are initially randomly oriented, only ~50% are modified by the hydrolase in the absence of flippase. In the presence of flippase, up to 100% of LLOs can be modified because of transfer between the inner and outer leaflets. This assay confirms the flippase activity of Rft1 and an archaeal homolog.

Finally, the authors address the original biochemical experiments threw doubt on the flipase function of Rft1. By reproducing these experiments in a more rigorous fashion and using the new flippase assay, they demonstrate the complete absence of flippase activity in membrane fractions prepared from $\Delta rft1$ yeast.

The experiments are rigorous and of high quality. The manuscript is clear, concise and well-structured. The conclusions are strongly supported by the data presented.

It is a pleasure to recommend this manuscript for publication. I don't see more than the following typos that need to be corrected..

- Page 3, line 48, "four Man and 3 Glc" please write out all small numbers in the text.

- Page 5, line 100, The term "size chromatography" is incorrect. Hydrophilic interaction chromatography was used to separate the glycans.

Sincerely,
Tim Keys

Reviewer #3 (Remarks to the Author):

In "Rft1 is necessary and sufficient for ER lipid-linked oligosaccharide translocation," Shuai Chen et. al. describe their results analyzing the role of Rft1p in the N-linked oligosaccharide pathway. They show that RFT1 orthologues from humans, trypanosomes and euryarchaeota can complement scRFT1 in yeast. Via a suppression screen they also identify a more distant orthologue in Agl23 in *H. hispanica*. Finally, the authors developed an in vitro assay for the flipping of M5GN2-PP-Dol. In this assay Rft1 is required for M5GN2-PP-Dol flipping.

The work experimental data to confirm the homologs of RFT1 and assays provides the first in vitro evidence confirming that Rft1p is directly required for M5GN2-PP-Dol flipping.

The work is of high standard and well documented. There is however one issue that concerns me. The authors main claim is that they identified RFT1 as the M5GN2-PP-Dol translocase. There is no evidence of such. In the contrary, RFT1 was identified in a publication they cite (28). As an author of that publication, I am shocked by how badly it is misrepresented here. All claims to have identified the flippase must be removed.

Regarding the value of the in vitro flipping assays, these nicely confirm the essentiality of Rft1p for M5GN2-PP-Dol translocation and provide evidence contrary to two previous attempts at using in vitro assays to assay M5GN2-PP-Dol activity.

Even without the main claim of the authors, the manuscript is of sufficient interest. Lipid translocases are essential for the functions of and on cellular membranes. They remain difficult to study and all advances in the methods used are of importance.

very minor note:

A plating assay is not sufficient to claim near-normal growth rate.

Response to reviewers

We would like to thank the reviewers for their valuable comments on our manuscript entitled "Rft1 is necessary and sufficient for ER lipid-linked oligosaccharide translocation" submitted to Nature Communications (Manuscript ID: NCOMMS-23-36106-T). Our changes, including the modifications and corrections requested from reviewers, are highlighted in the revised manuscript (emphasized with yellow background).

REVIEWER #1(Remarks to the Author):

Reviewer's comment: (written in blue)

This is a provocative, potentially important paper concerning the function of Rft1, an essential membrane protein in yeast and humans. A yeast genetics study in 2002 suggested that Rft1 flips the glycolipid Man5GlcNAc2-PP-dolichol across the ER membrane, a critical step in the assembly of the oligosaccharide precursor necessary for protein N-glycosylation. Subsequent biochemical studies indicated that this was not the case. Of relevance to the current paper, some of these studies recapitulated Man5GlcNAc2-PP-dolichol flipping in vesicles reconstituted with ER membrane proteins from yeast or rat liver and showed - using multiple techniques - that the flipping activity could be convincingly resolved from Rft1. These studies concluded that Rft1 is not necessary for flipping Man5GlcNAc2-PP-dolichol. These studies also showed that flipping is highly specific for Man5GlcNAc2-PP-dolichol, and very fast, occurring on the order of 1 minute.

The present paper claims that Rft1 is indeed the Man5GlcNAc2-PP-dolichol flippase by (i) identifying an archaeal protein (Agl23), a putative hexose-P-dolichol flippase, that can rescue Rft1 deficiency in cells, (ii) demonstrating the flippase activity of both Rft1 and Agl23 after reconstitution into vesicles, and (iii) using an enigmatic strain lacking Rft1 but over-expressing Hrd3, which retains viability but does not appear to have Man5GlcNAc2-PP-dolichol flippase activity when tested via biochemical reconstitution. There are numerous technical and conceptual problems with the paper.

Detailed comments are provided below. The main issues concern the authenticity of the purified proteins used for reconstitution and assay, lack of specificity controls to validate the assay, incomplete description of methods, and errors in describing previous work. The latter point derails the authors attempt to reconcile their results with previous reports.

Response:

We are very appreciative of this reviewer's suggestions. Their main issues concern (1) the authenticity of the purified proteins used for reconstitution and assay; (2) the lack of specificity controls to validate the assay; (3) an incomplete description of methods, and perceived errors in describing previous work. All of these concerns have been carefully considered and corrected in the revised manuscript, both experimentally and by textual revisions. Our point-by-point responses to comments are listed below:

Reviewer's comment:

Point 1. “Figure 1d and 3c. The oligosaccharide profiles are strange. Only GlcNAc₂ is seen in WT (and rescued) cells, whereas a more complete complement of LLOs would be expected, especially Man₅GlcNAc₂, Man₉GlcNAc₂, and Glc₃Man₉GlcNAc₂. Indeed, it is surprising to see GlcNAc₂ at all, as GlcNAc₂-PP-dol would be expected to be removed in the less polar solvent (CM 3:2) used for initial extractions. These points need to be addressed. Also relative abundance is not an adequate readout here as it should be demonstrated more exactly how much Man₅GlcNAc₂-PP-dol accumulation in the glucose-grown GAL-Rft1 cells compared with WT.”

Response:

The accumulation of M₅Gn₂-PP-Dol in Rft1-depleted yeast relative to wild type strains has been previously established. The experiments performed in our study shown in Figure 1d and Figure 3c were designed to compare the levels of LLO intermediates, particularly M₅Gn₂-PP-Dol, in Rft1-repressed (*GALpr-RFT1*) and *rft1Δ* strains as a function of the various suppressor genes they harbor, i.e. *HsRFT1*, *HhAGL23* and *ScHRD3*. The UPLC-MS based assay used for analyzing GlcNAc₂ LLOs in this study cannot be quantitatively compared to assays used in previous studies for the following reasons:

- (1) Previous methods used for measuring LLOs rely on isotopic labeling with [³H]-mannose. Therefore, only those LLOs containing mannose can be labelled with ³H and detected. Thus, GlcNAc₂- would have been missed in those previous analyses. Our UPLC-MS based assay revealed a large amount of GlcNAc₂- intermediate in wild-type cells. This is a novel observation that has not been reported, made possible by the uniqueness of the UPLC-MS assay system.
- (2) We detected the M₅GlcNAc₂- intermediate from the LLO fraction extracted from the wild type cells, but the relative abundance was very low compared to GlcNAc₂- (see **the Figure 1d in revised manuscript**). Moreover, we also observed that the relative amount of GlcNAc₂- decreases concomitantly with M₅Gn₂- accumulation in the glucose-repressed *GALpr-RFT1* strain (see **the Figure below**). This increased amount of M₅Gn₂- in glucose-repressed *GALpr-RFT1* cells (see revised Figure 1d or the figure below) is consistent with the phenotype reported by previous studies. Furthermore, the accumulation of M₅GlcNAc₂- in *rft1Δ* cells is eliminated by expression of *RFT1* orthologues or *Agl23* (Figure 1d), but not by overexpression of *HRD3* (Figure 3c). These phenotypes parallel both growth rate and CPYSDS-PAGE mobility in these strains (see Figure 1b & 1c and Figure 3a & 3b), further supporting the validity of our assay that monitors LLO accumulation in *rft1* deficient cells.
- (3) While our UPLC-MS-based assay has the advantage of avoiding the use of hazardous isotopes, a disadvantage is that the large molecular weight of the M₉GlcNAc₂- and G₃M₉GlcNAc₂- LLO intermediates diminishes the efficiency of their detection by mass spectrometry (MS). Nevertheless, we believe that in this study our UPLC-MS assay is sufficiently sensitive for measuring the variation of M₅GlcNAc₂- intermediate as a function of Rft1 depletion.
- (4) The reviewer was concerned about the use of the chloroform/methanol (CM) 3:2 solvent used for initial extraction can remove the GlcNAc₂-PP-Dol from the LLO fraction. We understand reviewer's concern. While there may be a better ratio of C/M to optimize extraction of GlcNAc₂-PP-Dol, our protocol enabled detection GlcNAc₂- from our LLOs extraction. Thus, any loss caused

by this CM 3:2 solvent is limited. Most importantly, we intentionally chose this method for extracting and monitoring the M5GlcNAc2-PP-Dol intermediate rather than GlcNAc2-, as this was the experimental goal.

In the revised version, more **detailed descriptions** have been added to explain results of Figure 1d.

Detection of M5GlcNAc2- intermediate from Rft1-depleted yeast

The W303a-*GALpr-RFT1* strain was used in this experiment. After culturing to 1-3 OD₆₀₀ in galactose medium, yeast cells were harvested and shifted to glucose-containing medium to repress *RFT1*. LLO intermediates were extracted from *RFT1*-repressed cells at 0, 6 and 12 h after shifting to glucose, prepared, and analyzed by UPLC-MS as described in Materials and Methods. Major peaks corresponding to GlcNAc2 and Man5GlcNAc2 are marked. The percentage of total signal in marked peaks is indicated.

Reviewer's comment:

Point 2. “Figure 2c. “It is the experience of this reviewer that Rft1 runs anomalously fast on SDS-PAGE gels, below the 50 kDa marker. The bands in the figure corresponding to FLAG-tagged ScRft1 and HsRft1 are in the 60-70 kDa range, at the size predicted from sequence, but not corresponding to how the protein behaves on SDS-PAGE. There is something critically wrong here and this point must be resolved. It is also very surprising to learn that the authors lyophilized their protein preps (lines 380 and 388) - please clarify.”

Response:

We agree and appreciate your professional suggestion. To address these concerns, we performed peptide sequence analysis of the Rft1 band after SDS-PAGE to verify its identity using non-lyophilized ScRft1. Additionally, to eliminate any doubt after purification, we re-did all assays using non-lyophilized purified ScRft1.

FLAG-ScRFT was expressed in W303 cells, purified as described, analyzed by SDS-PAGE and immunoblotted with FLAG-specific antibodies. Consistent with our previous results (shown in Figure 2c), both SDS-PAGE and immunoblotting analysis reproducibly detected FLAG-ScRft1 at the 50-70 kDa range (see **Figure a below**).

Furthermore, the band corresponding to FLAG-ScRft1 was cut out of the gel, digested, and subjected to peptide sequence analysis using LC-MS/MS analysis. As shown in Figure b and c below, peptides of ScRft1 were unambiguously identified, demonstrating that the band at 50-70 kDa does indeed correspond to ScRft1. For the above reasons, Figure 2c in the manuscript will **remain the same**.

C

FLAG-ScRft1:
 MDYKDDDDKAK**KNSQLPSTSEQILER**STTGATFLMMGQLFTK**LVTFILNLLIR**FLSPRIFGITAF
 LEFIQGTVLFFSRDAIRLSTLR**ISDSNGIIDDDEEYQETHYK**SKVLQAVNFAYIPFVWTFPLS
 IGLIAWQYRNINAYFITLPFFRWSIFLIWLSIIVELLSEPPFIVNQFMLNYAARSRFESIAVTTGCIVN
 FIVVYAVQQSRYPMGVVTSIDKEGIALAFALGKLAHSITLLACYWYDYLKNFKPKKLFSTRITK
 IKTRENNELKKGYPK**STSYFFQNDILQHFKKVYFQLCFKHLLEGGDK**LIINSLCTVEEQGIYALLS
 NYGSLLTR**LLFAPIEESLR**LFARLLSSHNPKNLKLSIEVLVNLTRFYIYLSLMIIVFGPANSSFLLQ
 FLIGSK**WSTTSVLDTR**IVYCFYIPFLSLNGIFEAFFQSVATGDQILKHSYFMMAFSGIFLLNSWLLI
 EKLKLSIEGLLSNIINMVLRLIYCGVFLNKFH**RELFTDSSFFNFK**DFKTVIAGSTICLLDWWFIG
 YVKNLQQFVVNVLFAMGLLALILVKER**QTIQSFIN**KRAVSNSKDV*

Figure for addressing the SDS-PAGE analysis of Rft1 protein

Recombinant FLAG-tagged ScRft1 protein was expressed in W303 cells and purified. (a) *top panel*:

Coomassie stained SDS-PAGE of total proteins and purified FLAG-ScRft1; *bottom panel*: Western blot of total proteins and purified FLAG-ScRft1 with anti-FLAG antibody. (b) The band marked in the red box was cut out and enzymatically hydrolyzed into peptides followed by LC-MS/MS analysis. As an example, secondary spectra of two peptide segments are represented. (c) Matched peptides of Rft1 are shown in red.

Reviewer's comments:

Point 3. “Figure 2d. “The kinetics of mannosidase treatment are very slow, with a half-time of 30 min judging from the liposome data. For vesicles reconstituted with ScRft1 and Agl23, the half-time of conversion of M5 to M3 appears to be about 1 hr. If the kinetics of mannosidase treatment and flipping are indeed resolved as suggested by these numbers (flipping being slower than mannosidase probing), then flipping in this assay is much slower than previously reported (line 54). The authors should address this issue.”

Responses:

Lipid movement across the membrane bilayers happens rapidly. All flippase activities (including the M5-PP-Dol flipping) that have previously been reported are dependent on the assays used for readout. For instance, in the previous M5GN2-PP-Dol assay, ConA capturing reflects the M5 flippase activity. In this study, the alpha-1,2 mannosidase activity reflects the M5-PP-Dol flipping. Although the kinetics of mannosidase treatment is much slower than the M5GN2-PP-Dol flipping, it can nevertheless reliably monitor the M5GN2-PP-Dol flipping across the membrane of liposome. However, it is not appropriate to compare the half-time of mannosidase activities in liposomes with “non” flippase activity (such as protein-free and HhAgl22-containing liposomes in Figure 2) versus those liposomes reconstituted with Rft1 or Agl23 since mannosidase treatment in these two kinds of liposomes represent different reaction processes.

In liposomes with non flippase activity, alpha-1,2 mannosidase can only access **50%** of the total M5Gn2-PP-Phy (M5), the population that is localized in the outer membrane leaflet and convert them to M3Gn2 (M3). As shown in Figure 2d, at 30 min, about half of the outer leaflet M5 was converted to M3. Thus, 30 min is the half time of mannosidase treatment under these conditions. However, this half of converted M5 can only be counted as **25% of total M5 substrate** (see Figure 2d).

In the case of liposomes reconstituted with Rft1 or Agl23, the mannosidase also accesses 50% of total M5 at the starting point. However, because the M5 can continuously flip across the membrane of liposomes, the luminal M5 will be exposed to the mannosidase and eventually become the substrate for the mannosidase. As shown in Figure 2d, at the timepoint of 1hr (60 min), about **50% of total M5 substrate** was converted to M3, resulting an extended half time for mannosidase treatment.

As described above, our M5GN2-PP-Dol assay is two-enzyme (flippase and alpha-1,2 mannosidase) system. We would like to emphasize again that, because the M5Gn2-PP-Dol flipping happens rapidly, the alpha-1,2 mannosidase activity reflects the M5 flipping activity in our assay system.

Reviewer's comments:

Point 4. “Figure 2e. Reconstitution of a purified protein using the Triton/BioBead protocol of Rigaud (ref 36) results in unilamellar vesicles in the 150-200 nm size range. Protein occupancy during reconstitution is dictated by Poisson statistics. For a protein of the size of Rft1, Poisson analysis

indicates that at a PPR of ~0.2 mg/mmol, 63% of the vesicles should have 1 or more flippases. Inspection of the data in Figure 2e indicates that the authors need to go to 15x higher PPR (3 mg/mmol) to achieve this level of occupancy. This is an important issue that the authors need to address. Also: how was the PPR determined (line 406 cites REF without providing a reference) and what were the protein and lipid recoveries after reconstitution?”

Responses:

In our original study, the PPR was calculated directly using the number of phospholipids and proteins added. This method ignored the degradation of phospholipids and proteins during liposome construction, resulting a higher PPR. To address the reviewer’s concern and ensure the accuracy and reliability of our flipping assay, we re-prepared proteoliposomes using purified ScRft1, **prepared without lyophilization**, and calculated its PPR.

The method we used for calculating the PPR of proteoliposomes was as described by Birgit Ploier and Anant K. Menon (doi:10.3791/54635), in which the phospholipid recovery was determined by a phosphate assay, and the protein recovery was estimated by quantitative western blot analysis. Our results confirmed that the phospholipid recovery rate of protein-free liposomes is $61.50 \pm 3.09\%$, while ScRft1-proteoliposomes revealed $59.54 \pm 1.73\%$ phospholipid recovery and $39.54 \pm 2.6\%$ protein recovery. In accordance with the above recoveries, we chose ScRft1- proteoliposomes with PPR of ~2.0 mg/mmol for re-analyzing the time course of translocation (i.e. M5GN2-PP-Phy hydrolysis by α -1,2 mannosidase). The new time course was **updated in Figure 2d** (see the revised manuscript). In addition, translocation rates as a function of varying PPR for 2 h were also re-calculated and **updated in Figure 2e** (see the revised manuscript).

Reviewer’s comments:

Point 5. “Figure 2d-f. There are no specificity tests to know if the activity that the authors describe matches their recapitulation of published reports that flipping has 'exquisite substrate specificity' (line 54-55). This would be an important validation of the assay being used and would also address the issue of whether Agl23 - a putative Hex-P-Dol transporter which would not have been expected to recognize a pyrophosphate-containing lipid - has similar specificity in this assay when compared with ScRft1 or HsRft1. Following the approach described in ref 26, the authors could therefore test other oligosaccharide-PP-dol species that are susceptible to the mannosidase used in their assay. They could also test structurally unrelated lipids such as glycerophospholipids using standard assays (for example, described by Wang et al. for CLPTM1L).”

Responses:

We tested the flipping activity of ScRft1 using **M3GN2-PP-Phy** as the substrate. The resulting evidence have been added into the manuscript as **Figure 4** (see the revised version).

In vivo, the M5GN2-PP-Dol flippase translocates the M5GN2 intermediate across the ER membrane from the cytoplasmic side into the lumen. As a physiologically relevant test for flippase substrate specificity, we used the M3GN2 intermediate as this represents another LLO intermediate normally found on the cytosolic face of the ER, versus other oligosaccharide-PP-Dol species such as M6GN2 or M9GN2 intermediates, which are located on the luminal side of ER membrane.

As presented in the Figure 4 of the revised manuscript, the ScRft1 failed to flip M3GN2 intermediate across the membrane, demonstrating its strong selectivity of M5GN2 over M3GN2, and suggesting selectivity as a LLO flippase for ER LLO. On the other hand, Agl23 displayed robust M3GN2 translocating activity, validating its previously described flipping activity on pyrophosphate-containing glycolipids.

Reviewer's comments:

Point 6. “Extended Data Figure 4. The result that increased expression of HRD3 can rescue the loss of Rft1 via the Tet-off system is intriguing. The data documenting this are not clearly explained. Extended Data Figure 4b is not at all clear - the explanation needs to be written out properly and a better figure needs to be presented. No sequence data are provided to support this result. It is surprising that despite switching to a TEF promoter, HRD message levels are only increased 1.5-fold. A protein blot should be shown to see exactly the increase in level of HRD3 protein that rescues the Tet-off RFT1 strain when grown on Dox. Can HRD overexpression from a plasmid rescue glucose-grown PGAL-RFT1 cells?”

Responses:

In the first place, we would like to emphasize that the main yeast strain tested in Figure 3 was the HRD3 high copy *rft1Δ* suppressor strain (*rft1Δ*+HRD3), in which HRD3 was overexpressed in a *rft1Δ* strain. The reviewer may have misread this experiment. We apologize that Extended Data Figure 4 was not clearly explained and therefore caused this misunderstanding. We also thank the reviewer very much for pointing this lack of clarity. In the revised version, more detailed explanations have been added to the methods section and the legend of Extended Data Figure 4 for documenting our experiments.

The suppressor screen described in Extended Data Figure 4 aimed to explain the idea why we constructed this *rft1Δ*+HRD3 strain and how we serendipitously discovered that overexpression of HRD3 rescues the growth of *rft1Δ* cells. Comments and concerns from the reviewer only focus on experiments performed in Extended Data Figure 4. None of them directly bear on the quality of data presented in Figure 3. Our responses addressing reviewer's suggestion and concerns are described below:

As shown in Extended Data Figure 4c, quantitative PCR analysis revealed a 1.5-fold increased HRD3 message RNA level in the *hrd3::PB P_{tet-off}-RFT1* mutant (see Extended Data Figure 4a & 4b) compared to the *ade2::PB P_{tet-off}-RFT1* strain (see Extended Data Figure 4a). The reviewer was concerned as they expected a higher-fold of increase of HRD3 mRNA since the TEF promoter is a strong constitutive promoter. However, these are the data and we nevertheless followed up to further test independently if overexpression of HRD3 could rescue the growth of *rft1Δ* cells. And in fact, we found that when expressed in another high copy plasmid (YEp351-GAPII-HRD3), overexpression of HRD3 did rescue the growth defect of *rft1Δ* cells but not its glycosylation defect (Figure 3a).

Moreover, the sequencing data proving the insertion site of the PB element was also submitted.

Reviewer's comments:

Point 7. “Figure 3b. The cells being used are presumably Tet-off RFT1 grown on Dox, and not *rft1*-delta as used in Figure 1 - the authors should make this clear. HRD over-expressing cells show hypoglycosylated CPY - this is true, but it is also clear that CPY is indeed glycosylated with bands corresponding to 1, 2, 3 and 4 N-glycans. The authors claim (based on panel d) that this strain has no Man5GlcNAc2-PP-dol flippase activity and yet CPY receives N-glycans! This is an important paradox that the authors need to address.”

Responses:

As we indicated in our above response to the reviewer, the strain used in Figure 3, in which *HRD3* was overexpressed by a YEp351-GAPII-*HRD3* plasmid, is in a *rft1Δ* background. We added several sentences in the revised version to emphasize this point.

As described in the manuscript, this *rft1Δ+HRD3* strain has a near-normal growth but accumulates M5GN2, suggesting it is specifically defective in M5GN2-PP-Dol luminal translocation (Fig. 3c). These findings were supported by the M5GN2-PP-Dol flipping assay using the membrane fraction of *rft1Δ+HRD3* cells (Fig. 3b). Our results suggested that *Hrd3* suppression of *rft1Δ* lethality involves a mechanism that bypasses the *rft1Δ*-dependent N-glycosylation defect. Although the basis for *Hrd3* suppression of *rft1Δ* is not yet understood, it is conceivable that a compensation mechanism exists in the *rft1Δ* strain, which can be triggered by the overexpression of *Hrd3* protein under the absence of *Rft1* background. Such a compensation mechanism may explain why CPY in the *rft1Δ+HRD3* cells receives a weak N-glycosylation.

Reviewer’s comments:

Point 8. “Figure 3d. This panel describes an assay done on vesicles reconstituted with a membrane protein extract. How was this extract prepared, characterized, and quantified? What PPR was used in the reconstitution?”

Responses:

To provide more details, we added the following text in the Method section of the manuscript:

Extraction of membrane proteins

Yeast cells were grown in YPAD to OD₆₆₀ ≈ 1.0. Yeast harvested from 200 mL were suspended in 5 mL lysis buffer [25mM HEPES pH 7.4, 150 mM NaCl and 1 × protease inhibitor cocktail (Roche/Mannheim, Germany)], and lysed by vortexing with glass beads on ice. The homogenate was centrifuged at 1,000 × g, 4 °C for 5 minutes to remove unbroken cells and cell fragments, 3000 × g, 4 °C for 5 minutes to further remove cell fragments and nuclear components, and 20,000 × g, 4 °C for 90 minutes to collect the membrane component. After washing the precipitate with 5 mL lysis buffer for three times, this membrane fraction was resuspended into 5 mL lysis buffer with 1% DDM, and kept at -80 °C. The concentration of membrane proteins was detected with BCA kit. The PPR of proteoliposomes reconstituted with these membrane proteins was ~20 mg/mmol.

Reviewer’s comments:

Point 9. Figure 3. As the authors are already preparing protein extracts and reconstituting them, it would help the controversy in this field if they could prepare an extract from FLAG-ScRft1-expressing yeast, and test activity after passing the extract over anti-FLAG resin to remove the tagged protein specifically. This would match the type of experiment previously reported in the literature, for example ref 24.

Response:

The reviewer suggested to test the M5GN2-PP-Dol flipping activity using a membrane extract from FLAG-ScRft1-expressing yeast after passing the extract over anti-FLAG resin to deplete the tagged protein specifically. However, we think it is unnecessary to perform this experiment and include in our manuscript for two reasons:

- (1) We already designed experiments to test the activity using the extract from the *rft1Δ+HRD3* cells, in which the *RFT1* gene was completely removed (Figure 3). This is the better method of ensuring the absence of Rft1.
- (2) The previous study used a ConA capturing M5-PP-Dol assay system, while our M5-PP-Dol assay is two-enzyme system (i.e. M5GN2-PP-Phy hydrolysis by α -1,2 mannosidase). These assays work with different molecular mechanisms and use different techniques. If no M5GN2-PP-Dol flipping activity were to be detected after passage of the extract over anti-FLAG resin, it would only demonstrate a complete removing of Rft1 from the membrane fraction by anti-FLAG resin. Nothing would help the controversy.

Reviewer's comments:

Point 10. “The authors make numerous errors in describing the previous biochemical data on Rft1.

(i) Many of the previous experiments were done on extracts from which Rft1 was eliminated (not simply depleted) - these extracts had full activity. For example, Ref 24 tested an extract from yeast cells expressing Rft1-Protein A in place of endogenous Rft1. Quantitative removal of this protein on IgG beads did not affect the Man5GlcNAc2-PP-dol flippase activity of the extract. Similarly, ref 23 fractionated an extract on blue dye resin and showed that the activity eluted at a medium salt concentration distinct from the Rft1 elution profile. Thus, the authors' repeated proposal (for example, lines 60, 161, 164-165, 203 and perhaps other places as well) that previous work was affected by incomplete removal of Rft1 is not correct.”

Response:

We thank the reviewer for pointing these out. It should be considered that, in the previous studies on M5GN2-PP-Dol flipping activity, proteoliposomes reconstituted with an extract on blue dye resin from the detergent-extracted yeast microsomal proteins were also used (doi:10.1021/bi800723n). Thus, following the suggestion by reviewer, we have modified the documents at Lines 60,161-165 and 203 of manuscript (see the revised manuscript). Moreover, the sentence describing the incomplete removal of Rft1 protein at Lines 164-165 was deleted.

(ii) Of note, some of the previous biochemical work was done on membranes or extracts prepared from glucose-grown cells expressing Rft1 under control of a GAL promoter. Even here, readouts such as the protein-dependence plots shown in Ref 24, are sensitive to the amount of remaining Rft1 – the

shape of the plots would have changed in a predictable way, yet no effect was observed. The authors need to consider this point carefully as it does not offer the 'out' that they propose to resolve the discrepancy between their data and previous reports.

Response:

We completely agree with reviewer's opinion. In this study, we primarily used purified Rft1 and a membrane fraction from *rft1Δ+HRD3* cells to assay the M5GN2-PP-Dol flipping activity, while previous study used Rft1 protein fractions prepared from yeast cells, including the *GAL* promoter-*RFT1* strain or the rat liver. In addition, the M5GN2-PP-Dol flipping assay system used in previous studies and this study are also different, further indicating there is no direct comparison between the results from these two studies. So, we should not judge previous results using our results and vice versa. The controversy of M5GN2-PP-Dol flipping has existed for more than twenty years. There are numerous biochemical studies that have been done during this period. It is beyond the scope of the current study and indeed unfair to request that we repeat all those previous experiments to seek conformity. We are not in the position to judge those legendary studies using our results, which have been done with different assay system and different protein samples because we respect researchers who have performed those works. Here, we have strived to provide reproducible high-quality data to interpret.

(iii) It is interesting to note that Rft1 is itself not a glycoprotein. Yet a previous report (PMID 20534553; DOI: 10.1073/pnas.1002408107) shows that the protein(s) responsible for Man5GlcNAc2-PP-dol flippase activity bind to Con A Sepharose. This shows a completely different way of resolving Rft1 from Man5GlcNAc2-PP-dol flippase activity, unrelated to genetic depletion or immunodepletion protocols noted in points (i) and (ii).

(iv) Also of note, a recent paper (PMID 33446867) - which the authors should cite - used an orthogonal approach to identify Man5GlcNAc2-PP-dol flippase candidates. Several proteins were identified, but not Rft1 whose chromatography profile did not match that of the activity.

Response:

There are no reports in the literature that have experimentally determined whether or not Rft1 is glycosylated. To address this question, we treated ScRft1 with Endo H and found no evidence for sensitivity, suggesting the ScRft1 is not N-glycosylated. The first paper (PMID 2053455) cited by the reviewer suggested that the Man5GlcNAc2-PP-Dol flippase is itself or associated with a mannosylated protein. The second paper (PMID 33446867) cited by the reviewer postulated that Man5GlcNAc2-PP-Dol flippase activity is likely to be regulated by interaction with other molecules in a complex. While both studies concluded that Rft1 is not the Man5GlcNAc2-PP-dol flippase, neither study provided the identity of this putative flippase.

As researchers, we respect all the previous results and will cite and try to discuss them in our manuscript (see the Ref 41, 42). On the other hand, in this study, we are presenting a large amount of data that demonstrate purified ScRft1 or HsRft1 protein can translocate the M5GN2-PP-Dol or M5GN2-PP-Phy across the membrane, and the membrane fraction prepared from a *rft1* null strain fails to do so. It seems unreasonable to request that we perform additional experiments to demonstrate that Rft1 has attributes that are similar to a group of proteins with uncertain identity.

(v) Lines 204-205 state that previous experiments were done on liposomes reconstituted with proteins from an RFT1-null trypanosome strain. This is not correct. Similarly, lines 211-212 are incorrect.

Response:

We have incorrectly described the previous biochemical results representing phenotypes of the RFT1-null trypanosome strain and thank the reviewer for pointing these out. This text at lines 204-205 was removed and 211-212 and replaced with new text.

Other points

Reviewer's comments:

1. The first section of the results (line 72) initially gives the impression that a suppressor screen was carried out, whereas the authors simply picked a selection of candidates to test. Group I candidates are Rft1 orthologs, and all of these can functionally substitute for yeast Rft1 as expected. Although the authors did come up with Agl23 as a suppressor, the choice of Group II candidates is poorly justified as the Man5GlcNAc2-PP-dolichol flippase activity is known to be highly specific (line 55 and cited references). Thus, the flippase is unlikely to transport substrates with a monophosphate linkage (i.e., substrates of CLPTM1L and Agl23), or substrates with an unsaturated alpha-isoprene (substrate of MurJ) and so it is not clear why these proteins were chosen! CLPTM1L and MurJ do not complement the *rft1*-delta strain - but are these proteins expressed in yeast? if expressed, are they in the right compartment?

Response:

The reviewer indicated that “the choice of Group II candidates is poorly justified”. Theoretically, he is right. However, ‘poor’ doesn’t mean ‘wrong’. As we presented in Figure 1, we did pick up Agl23 as a protein that rescues the *rft1* deletion phenotype. We chose Agl23, CLPTM1L and MurJ as candidates for group II because they have potential glycolipid flipping activity. CLPTM1L is an ER lipid scramblase (doi.org/10.1073/pnas.2115083119); The bacteria MurJ protein belongs to the multidrug/oligosaccharyl-lipid/polysaccharide (MOP) exporter superfamily, in which Rft1 family is also included (doi:10.1046/j.1432-1033.2003.03418.x). In addition, MurJ and Rft1 share structural similarity. Before our test, their expression in yeast cells has been confirmed by western blot. Furthermore, their ER localization was also confirmed by membrane fractionation.

2. Figure 2. The data appear to be technical replicates - three measurements done on the same batch of liposomes. This is insufficient. Unique protein preps, and unique reconstitutions should be done to reproduce the data.

Response:

Data are from three independent measurements (Fig. 2d, e, f and Fig. 2d.). The relevant information has been included in the figure legends of the revised manuscript.

3. The description of proteoliposome reconstitution (line 390) is inadequate and makes little sense as

written. For example, lipids are dried, dissolved in detergent and then extruded (lines 395-397)? This would not result in liposomes! Were the proteoliposomes also extruded?

Response:

We have made corrections in the ‘Method’ section of revised manuscript as follow:

To make the proteoliposomes, 100 µl of buffer B (10 mM HEPES-NaOH, pH 7.4, 150 mM NaCl, 0.1% DDM) containing different proteins (as indicated) was mixed with prepared protein-free liposomes in buffer A and incubate at 4 °C for 1 hour with end over end rotation before adding Bio-beads.

4. The activity assay (line 408) needs to be written out more clearly. The samples are prepared in Hepes/NaCl, but then diluted into MES/CaCl₂ -for assay - this is strange. Why was the buffer changed?

Response:

We appreciate your suggestion. More detailed information describing the flipping assay has been added into the ‘Method’ section of revised manuscript.

Because we re-did the flipping assay using non-lyophilized for the revision, the Hepes/NaCl buffer was used for reaction, in which CaCl₂ was supplemented for the activity of mannosidase.

At the end of the assay, it appears that the samples are heated at 95C, then HCl is added, and the samples are further heated. Is this treatment sufficient to capture all glycans? Why not extract the lipids first (as in Fig 1d) and then treat with HCl to obtain glycans? Line 414 - oligosaccharides were purified - how?

Response:

These treatments have been previously reported (Zufferey, R. et al. STT3, a highly conserved protein required for yeast oligosaccharyltransferase activity in vitro. EMBO J. 14, 4949±4960 (1995)). Our study has proved that this treatment is good enough to release and detect glycans intermediates.

Oligosaccharides were purified as follow: ‘The released oligosaccharides were purified by solid-phase extraction using 1mL Supelclean ENVI-Carb Slurry (Sigma-Aldrich, MO, USA).’

The above description has been added into the ‘Method’ section of revised manuscript.

5. Figure 2d. Although Extended Data Fig 3 shows exemplary chromatograms related to mannosidase treatment of liposomes (what are the incubation times going from step II to II, and III to IV? this information is not provided in the legend), a full data set of UPLC chromatograms should be provided to accompany the traces in Fig. 2d. What fraction of the Man5GNPPlipid committed to the flippase experiment is recovered in the UPLC chromatograms as Man3GN2 and Man5GN2?

Responses:

We made modifications to Extended Data Fig 3 by adding incubation time. The incubation times from step II to II was 2 h, and III to IV was 1h. If necessary, Full data set of UPLC chromatograms corresponding to Fig. 1d, Fig. 2d, e, f, Fig. 3d, Extended Data Fig. 2b, c, e, f, and Extended Data Fig.

3b will be provided and uploaded to the database. In the chromatograms, Man3GN2 and Man5GN2 correspond to 933 m/z and 1257 m/z respectively.

6. Figure 2d,e,f: The y-axis label needs to be changed. "The ratio of M5-GN2 (%)" does not make sense. Presumably the authors mean $100 \times M5 / (M3 + M5)$?

Response:

We agree and appreciate your suggestion. The y-axis label indicates the proportion of M5-GN2 in the sum of M3-GN2 and M5-GN2. We made modifications to the figures in the manuscript.

7. A strain list needs to be provided and the authors need to be clear about what strains are used in each of the experiments. For example, note the ambiguity in point 7 (main comments).

Response:

Tables of strains and plasmids used in this work have been included in the revised manuscript.

8. The last line of the abstract is a little over the top! Even if the authors are correct in their conclusions, the flippases for Man-P-dol and Glc-P-dol, necessary for the conversion of Man5GlcNAc2-PP-dolichol to the canonical oligosaccharide donor Glc3Man9GlcNAc2-PP-dolichol, remain to be identified.

Response:

We are grateful for the opinion and suggestion. We revised the relevant text in the manuscript.

9. These points need to be corrected/addressed. Line 43: not all eukaryotes use a 14-sugar oligosaccharide. Line 46: sugars are added to Dol-P, not Dol-PP (-PP- is formed when GlcNAc-P is added to Dol-P as shown in Fig 1A). Line 55: replace 'Such a' with 'A putative'. Line 59: proteoliposomes reconstituted from detergent-solubilized yeast Er membrane proteins. Line 61: the eukaryote/protist Plasmodium, has an N-glycosylation pathway but lacks Rft1 - this point should be added. Line 85: GlcN-PI AND phospholipids Line 89: it would be useful to state here that Agl22 is a glycosyltransferase. Lines 166-167: 'Rft1 is strictly required ...' - this statement is not justified based on Fig 2, where Agl23 appears to perform the same function as Rft1 Line 197: this is not a screen, simply a test of candidates - the sentence should be modified. Line 224: how many hours in glucose? Line 238: is this a Coomassie-stained gel? Please specify. Coomassie-stained gel Line 365: what is 'oligosaccharide transfer rate'?

Response:

We agree and appreciate your meticulous suggestions. The relevant modifications are as follow:

Line 43: not all eukaryotes use a 14-sugar oligosaccharide.

‘Eukaryotes’ has been changed to ‘Most eukaryotes’

Line 46: sugars are added to Dol-P, not Dol-PP (-PP- is formed when GlcNAc-P is added to Dol-P as shown in Fig 1A).

'Dol-PP' has been changed to 'Dol-P'

Line 55: replace 'Such a' with 'A putative'.

This suggestion conflicts with the concern from the reviewer 3. **We leave the decision of this replacement to the editor.**

Line 59: proteoliposomes reconstituted from detergent-solubilized yeast ER membrane proteins.

The sentence has been changed to 'proteoliposomes reconstituted with detergent-solubilized ER membrane proteins from rat liver or yeast cells, in which Rft1 was reduced or removed nevertheless displayed robust flippase activity'

Line 61: the eukaryote/protist Plasmodium, has an N-glycosylation pathway but lacks Rft1 - this point should be added.

We appreciate this valuable suggestion. As the reviewer indicated, not all parasitic protozoans possess a *RFT1* homologue and have an N-glycosylation pathway like *T. brucei*. For instance, *Perkinsus marinus*, *Gregarina niphandrodes* and *Toxoplasma gondii* possess most of *ALG* (asparagine-linked glycosylation) homologues but lack *RFT1* (Lombard Biology Direct (2016) 11:36, DOI 10.1186/s13062-016-0137-2). We have added this point into the 'discussion' section of the revised manuscript.

Line 85: GlcN-PI AND phospholipids

It has been changed to 'GlcN-PI and phospholipids.'

Line 89: it would be useful to state here that Agl22 is a glycosyltransferase.

It has been indicated in the revision.

Lines 166-167: 'Rft1 is strictly required ...' - this statement is not justified based on Fig 2, where Agl23 appears to perform the same function as Rft1 \

'strictly' has been deleted.

Line 197: this is not a screen, simply a test of candidates - the sentence should be modified.

The sentence has been modified.

Line 224: how many hours in glucose?

Yeast cells were grown in galactose (Gal)- or glucose (Glc) for 12 h.

Line 238: is this a Coomassie-stained gel? Please specify. Coomassie-stained gel

Yes, this is a Coomassie-stained gel. It has been explained in the revision

Line 365: what is 'oligosaccharide transfer rate'?

It has been changed to 'The relative oligosaccharide content was quantified by calculating the peak intensity in LC-ESI-MS using Xcalibur.'

Reviewer #2 (Remarks to the Author):

Reviewer's comments: (written in blue)

The study by Chen and colleagues provides a clear answer to a long standing and important biochemical question: The possible role of Rft1 in flipping the lipid-linked oligosaccharide (LLO) intermediate from the cytoplasmic to the luminal side of the ER membrane during synthesis of the donor substrate for N-linked glycosylation of proteins in eukaryotes.

The authors elegantly combine genetic screens with a novel biochemical flipping assay to demonstrate unequivocally that Rft1 is the flippase – as was reported by Helenius and Aebi based on genetic studies over 20 years ago and subsequently brought into question by biochemical studies. Using genetic screens they identify archaeal proteins that suppress lethality of a $\Delta rft1$ mutation in yeast. They proceed to develop a flippase assay that uses glycoside hydrolyse activity to modify LLOs that are exposed on the external surface of reconstituted liposomes. Because the LLOs are initially randomly oriented, only ~50% are modified by the hydrolase in the absence of flippase. In the presence of flippase, up to 100% of LLOs can be modified because of transfer between the inner and outer leaflets. This assay confirms the flippase activity of Rft1 and an archaeal homolog.

Finally, the authors address the original biochemical experiments threw doubt on the flippase function of Rft1. By reproducing these experiments in a more rigorous fashion and using the new flippase assay, they demonstrate the complete absence of flippase activity in membrane fractions prepared from $\Delta rft1$ yeast.

The experiments are rigorous and of high quality. The manuscript is clear, concise and well-structured. The conclusions are strongly supported by the data presented.

It is a pleasure to recommend this manuscript for publication. I don't see more than the following typos that need to be corrected.

- Page 3, line 48, "four Man and 3 Glc" please write out all small numbers in the text.
- Page 5, line 100, The term "size chromatography" is incorrect. Hydrophilic interaction chromatography was used to separate the glycans.

Sincerely,
Tim Keys

Response:

We appreciate and thank you for your review. The relevant modifications are as follow:

- (1) 'four Man and 3 Glc' has been changed to 'four Man and three Glc'
- (2) The oligosaccharides of LLO intermediates were analyzed by UPLC-MS with an Acquity UPLC BEH Amide column. We have modified the sentence as 'The oligosaccharides of LLO intermediates were analyzed by UPLC-MS confirmed that in these *rft1*Δ suppressed strains.'

Reviewer #3 (Remarks to the Author):

Reviewer's comments: (written in blue)

In "Rft1 is necessary and sufficient for ER lipid-linked oligosaccharide translocation," Shuai Chen et al. describe their results analyzing the role of Rft1p in the N-linked oligosaccharide pathway. They show that RFT1 orthologues from humans, trypanosomes and euryarchaeota can complement scRFT1 in yeast. Via a suppression screen they also identify a more distant orthologue in Agl23 in *H. hispanica*. Finally, the authors developed an *in vitro* assay for the flipping of M5GN2-PP-Dol. In this assay Rft1 is required for M5GN2-PP-Dol flipping.

The work experimental data to confirm the homologs of RFT1 and assays provides the first *in vitro* evidences confirming that Rft1p is directly required for M5GN2-PP-Dol flipping.

The work is of high standard and well documented. There is however one issue that concerns me. The authors main claim is that they identified RFT1 as the M5GN2-PP-Dol translocase. There is no evidence of such. In the contrary, RFT1 was identified in a publication they cite (28). As an author of that publication, I am shocked by how badly it is misrepresented here. All claims to have identified the flippase must be removed.

Response:

We sincerely apologize for any misrepresentation and attempted to clearly attribute the identification of Rft1 by elegant genetic and biochemical analyses to the original Helenius *et al* paper. We've checked our manuscript and found that in the last sentence of the Abstract (**Copy from the Abstract: With the identity of Rft1 as the M5GN2-PP-Dol flippase, we now know every enzyme in the N-linked lipid-linked oligosaccharide biosynthetic pathway.**), 'With the **identity** of Rft1 as the M5GN2-PP-Dol flippase,' is not appropriate and misrepresented the previous work done by Helenius et al. This sentence has been removed and we aimed to be clear that our work simply confirmed those previous findings. We cannot find any others that misrepresented the previous genetic study on identification of Rft1. If there are still such sentences or words, please point them out. We would like to remove or correct all of them.

Reviewer's comments:

Regarding the value of the *in vitro* flipping assays, these nicely confirm the essentiality of Rft1p for M5GN2-PP-Dol translocation and provide evidence contrary to two previous attempts at using *in vitro* assays to assay M5GN2-PP-Dol activity.

Even without the main claim of the authors, the manuscript is of sufficient interest. Lipid translocases are essential for the functions of and on cellular membranes. They remain difficult to study and all advances in the methods used are of importance.

very minor note:

A plating assay is not sufficient to claim near-normal growth rate.

Response:

Thank the reviewer to point this out. To precisely describe the growth rate of W303a-*rft1* Δ +*HRD3* strain, its time-dependent growth rate was calculated and compared with that of W303-*rft1* Δ +*ScRFT1* cells. The resulting growth curves are added to Figure 3a (see the revised Figure 3).

Reviewer #1 (Remarks to the Author):

The paper has been improved in several aspects and remains intriguing. However, the conclusions should be qualified/dialed down as they are not uniformly supported by the data. For the same reason, the title should be less definitive - perhaps 'Role of Rft1 in ER LLO translocation'. I have several questions which require significant changes to the text.

Point 1.

New data to test flipping of M3Gn2-PP-Dol (Figure 4).

(a) The data quality is not great. Given the slope of the graph it would appear that the lipid is being flipped at a slow rate even in liposomes as conversion of M3 to M2Gn2-PP-Dol would appear to continue past the 4 h timepoint, for both protein-free and ScRft1-proteoliposomes.

(b) A figure similar to extended Data Fig 3 should be provided to support this assay.

(c) The authors conclude from this new data set that ScRft1 does not flip M3Gn2-PP-Dol (but see above about the quality of the data). They contrast these data with a previous report where crude ER extracts from rat liver were reconstituted and tested - however, in these previous studies flipping of M3 was slower than that of M5, but because the M5 flip rate was over-estimated as it could not be resolved from the rate of probing, a precise fold-difference between M3 and M5 flipping could not be calculated, but was likely to be $\gg 2$ -fold). However, of considerable relevance is the result of Helenius et al. 2002 paper, who showed that the poorly growing alg11 mutant, which is blocked at the M3 stage and requires sorbitol to support growth, could be rescued by expression of Rft1. Helenius et al. concluded that M3 was a poor substrate for the flippase, but that over-expression of the putative flippase (ScRft1) could overcome the problem. This result is at odds with the result shown in Fig. 4b. The authors need to discuss this point in detail.

Point 2.

HRD3 data (Figure 3).

These data are intriguing and confounding. Expression of HRD3 from a strong constitutive promoter (Fig 3a), rescues rft1-delta cells. Thus, the cells have no Rft1, yet are able to grow almost as well as wild-type cells.

How do the authors reconcile the following three statements corresponding to their data?

(a) HRD3 cells are able to glycosylate CPY, albeit poorly. Even so, fully glycosylated CPY is present (Fig 3b), as well as several glycoforms, suggestive of an intact LLO pathway which necessarily includes flipping of M5 lipid.

(b) M5 lipid does not accumulate to the same extent as it does in GALpr-RFT1 cells grown on glucose (Fig. 1d). This indicates that M5 is being consumed, perhaps slowly. Presumably it is being flipped and used for LLO biosynthesis.

(c) Reconstitution of a protein extract from the cells shows no M5 flippase activity.

The authors state that Rft1 is necessary for flipping as they detect no flipping in their assay of a reconstituted HRD3 cell extract. Yet the cell-based data indicate that M5 must be flipped. This point requires detailed attention.

Point 3.

Flippase assay kinetics (Figure 2).

As the authors state in the rebuttal document, the readout of the M5 flippase assay (Fig. 2) depends on a combination of flippase activity and mannosidase activity and suggest that the mannosidase step is rate-limiting. The half-time of the mannosidase step is 30 min, and this should be the same for the flippase-containing vesicles if the mannosidase is rate-limiting. For example, inspection of Fig 3 in ref 35 (Huang et al. 2021) shows that the rate of fluorescence reduction is the same in protein-free vesicles and in TMEM41B proteoliposomes in an assay where NBD-PC is the lipid (=M5 in the current

assay) and dithionite is the probe (=mannosidase in this assay). I don't think that the analysis of the kinetics presented in the rebuttal document is correct - flipping seems to occur more slowly than the mannosidase reaction. This should be discussed properly in the text.

Point 5. Technical issues.

- (a) Line 486 - what mole percentage of M5 (or M3) lipid was used in the reconstitution?
- (b) Line 488 - it seems that the authors dried lipids, dissolved in buffer with 1% TX-100 and then inexplicably extruded the detergent-solubilized lipids 30x through a 400-nm filter. Why was the extrusion done?
- (c) Protein and lipid recovery after reconstitution (provided in the rebuttal document) should be included in the paper.
- (d) Fig 3a - is this an FOA plate? how was the empty vector control set up?
- (e) Fig 3a - why was the growth curve done in YAPD media? why not YPD?
- (f) Please include in Table 2 a description of the HRD3 plasmid used in Fig 3a.
- (g) Line 190 - the authors should add a quantitative comparison of the relative accumulation of M5 lipid in the HRD3 cells versus GALpr-Rft1 cells grown in glucose.
- (h) The rebuttal data on M5Gn2-PP-Dol accumulation, and concomitant decrease in Gn2-PP-Dol, as a function of time in glucose media of the GALpr-RFT1 strain, should be included as supplementary data, accompanied by a graph showing the relative change), as this would help convince the reader of the quantitative quality of this readout.
- (i) The authors should state in the text that purified ScRft1 was verified by LC-MS/MS sequencing and show, as a supplementary figure, the sequencing data presented in the rebuttal. Of note, the original comment about the incorrect size came from the labeling of molecular weight markers in Fig 2c - in the original manuscript the protein ran between markers labeled 60 and 70 kDa; in the revised manuscript, the gel remains the same, but the relevant markers are now 50 and 70 kDa. The gel figure (Fig 2c) has indeed been changed even though the authors say in their rebuttal that it remains the same.
- (j) Line 233 - add a phrase: HRD3 lacked this activity, yet glycosylated CPY.
- (k) Line 269 - please add Plasmodium.
- (l) Line 65 - modify as follows: ... displayed robust LLO-specific flippase activity.

Reviewer #3 (Remarks to the Author):

I thank the authors for their effort to correct claims of having identified Rft1p. I may be sensitive but with the statements "The identity of this translocator is an important unanswered question" and "Genetic studies implicate Rft1" in the abstract, the authors are suggesting that Rft1 was not identified.

REVIEWER COMMENTS

Reviewer #1 (Remarks to the Author):

Reviewer's comment: (written in blue)

The paper has been improved in several aspects and remains intriguing. However, the conclusions should be qualified/dialed down as they are not uniformly supported by the data. For the same reason, the title should be less definitive - perhaps 'Role of Rft1 in ER LLO translocation'. I have several questions which require significant changes to the text.

Response:

We thank the reviewer for their careful review of our manuscript and comments. We addressed the remaining points raised by the reviewer as much as possible and made revisions accordingly. Our changes, including the modifications and corrections requested from reviewers, are highlighted in the revised manuscript (emphasized with yellow background). In addition, according to the “guide for submission to Nature Communications”, we have renamed “**Extended Data Figures**” as “**Supplementary figures**”.

Point 1.

Reviewer's comment:

New data to test flipping of M3Gn2-PP-Dol (Figure 4).

(a) The data quality is not great. Given the slope of the graph it would appear that the lipid is being flipped at a slow rate even in liposomes as conversion of M3 to M2Gn2-PP-Dol would appear to continue past the 4 h timepoint, for both protein-free and ScRft1-proteoliposomes.

Response:

The graph presenting the M3 flipping reveals a slight slope between incubation time 2h and 4h, raising a question about whether or not spontaneous flipping of M3 might be occurring. We thank the reviewer for pointing this out. For a more accurate depiction, data at the 3-hour incubation time was included in the graph (see the revised Figure 4). These data demonstrate that the flipping kinetics of M3 in the presence of ScRft1 mirrors that of protein free-liposomes, which are markedly reduced from those with HhAgl23, where only 15% of the M3 remained.

Reviewer's comments:

(b) A figure similar to Supplementary Fig 3 should be provided to support this assay.

Response:

A modified figure similar to Supplementary Fig 3 has been added as **Supplementary Fig 7**, which validates this assay. Noticeably, the ratio of M3GN2 (M3) to M2GN2 (M2) is approximately 50% at the 4 h time point of α 1-2,3 mannosidase treatment (Supplementary Fig 7: step III), suggesting that the liposomes used in this assay

maintain their integrity to at least this time point of incubation.

Reviewer's comment:

(c) The authors conclude from this new data set that ScRft1 does not flip M3Gn2-PP-Dol (but see above about the quality of the data). They contrast these data with a previous report where crude ER extracts from rat liver were reconstituted and tested - however, in these previous studies flipping of M3 was slower than that of M5, but because the M5 flip rate was over-estimated as it could not be resolved from the rate of probing, a precise fold-difference between M3 and M5 flipping could not be calculated, but was likely to be $\gg 2$ -fold). However, of considerable relevance is the result of Helenius et al. 2002 paper, who showed that the poorly growing *alg11* mutant, which is blocked at the M3 stage and requires sorbitol to support growth, could be rescued by expression of Rft1. Helenius et al. concluded that M3 was a poor substrate for the flippase, but that over-expression of the putative flippase (ScRft1) could overcome the problem. This result is at odds with the result shown in Fig. 4b. The authors need to discuss this point in detail.

Response:

Helenius et al. (2002 paper) showed that overexpression of ScRft1 increases the use of M3 and partially suppresses the growth defect of *alg11*. This result supports the idea that when overexpressed Rft1 can flip M3 *in vivo*, but under normal levels of expression, it is a poor substrate. Our *in vitro* assay is an artificial system to measure the flippase activity, which is different from the *in vivo* situation. While we cannot rule out that our *in vitro* assay lacks the sensitivity to detect very low amounts of flipped M3, the main point is that Rft1 displays a strong preference for flipping M5 compared to M3 while *HhAgl23* does not. While it may be true that Rft1 flips M3GN2-PP-Dol *in vivo*, it does so less efficiently than M5. This idea (and the Helenius *et al.* results) have been included in the revision. (see Lines 233-235 in second revision)

Point 2.

Reviewer's comment:

HRD3 data (Figure 3). These data are intriguing and confounding. Expression of HRD3 from a strong constitutive promoter (Fig 3a), rescues *rft1*-delta cells. Thus, the cells have no Rft1, yet are able to grow almost as well as wild-type cells.

How do the authors reconcile the following three statements corresponding to their data?

Response:

There are three observations regarding *HRD3* suppression of *rft1* Δ that the reviewer queried (these are each addressed separately below). All our explanations regarding each of these queries are predicated on the idea that the essential function of *RFT1* is M5-PP-Dol flipping. Therefore, suppression of *rft1* Δ lethality is always accompanied by rescue of M5-PP-Dol flipping as these phenotypes are directly related to each other. All of our data support this hypothesis, including (i) archaeal *HhAgl23*, which bears no resemblance to Rft1, complements both the growth and glycosylation defects of *rft1* Δ cells. *HhAgl23* also displays robust M5 flipping activity *in vitro*. (ii) Ditto for Rft1 orthologues.

Reviewer's comment:

(a) HRD3 cells are able to glycosylate CPY, albeit poorly. Even so, fully glycosylated CPY is present (Fig 3b), as well as several glycoforms, suggestive of an intact LLO pathway which necessarily includes flipping of M5 lipid.

Response:

As the reviewer indicated, we observe some glycosylated forms of CPY in the *rft1Δ+HRD3* strain (Figure 3) but **glycosylation in this strain is severely impaired compared to wild type** (Fig.3b). Indeed, much of the observed CPY in *rft1Δ+HRD3* cells remains hypo-glycosylated (Figure 3b). Thus, the LLO pathway is not intact. Furthermore, accumulation of M5 is detected in this strain (Figure 3c). Therefore, our results demonstrate that although some M5-PP-Dol can cross the ER membrane, M5-PP-Dol flipping activity in *rft1Δ+HRD3* cells is impaired. We would argue that this weak *RFT1*-independent flipping of M5-PP-Dol (which occurs by an as yet undefined mechanism) is directly responsible for the viability of *rft1Δ+HRD3*. We have added more comments into the "Discussion" to emphasize this important phenotype of Rft1 protein (see Lines 265-294 in second revision).

As shown in Figure 3a, **the growth rate of *rft1Δ+HRD3* strain is also reduced compared to wild type** and we thank the reviewer for suggesting that more quantitative growth rates should be measured as these were required to quantitate this deficiency. To emphasize the difference of their growth phenotypes, we modified the text describing cell growth used at Line 188 in the revised manuscript from "This *rft1Δ+HRD3* strain had a near-normal growth rate" to "This strain, despite lacking RFT1, is able to grow though with a reduced rate compared to wild type (Fig. 3a). This strain also accumulates hypo-glycosylated CPY, indicating a defect in N-glycosylation (Fig. 3b)." (See the Line 198-203 in second revision).

Reviewer's comment:

(b) M5 lipid does not accumulate to the same extent as it does in GALpr-RFT1 cells grown on glucose (Fig. 1d). This indicates that M5 is being consumed, perhaps slowly. Presumably it is being flipped and used for LLO biosynthesis. The result in Fig. 3c indicate that less amount M5 is accumulated upon HRD3 overexpression. Fig.3b and Fig. 3c and Overexpression of HRD3 do cause some flipping of M5 lipid in *rft1*-delta cells (Fig. 3b).

Response:

Again, we agree with this comment and would argue that this supports the idea that overexpression of *HRD3* suppresses the lethality of *rft1Δ* by somehow enabling M5 (as well as M4, M3 M2...) flipping.

Reviewer's comment:

(c) Reconstitution of a protein extract from the cells shows no M5 flippase activity. The authors state that Rft1 is necessary for flipping as they detect no flipping in their assay of a reconstituted HRD3 cell extract. Yet the cell-based data indicate that M5 must be flipped. This point requires detailed attention.

Response:

Failure to detect *in vitro* M5 flipping activity using protein extracts from *rft1Δ+HRD3* cells reconstituted into artificial liposomes can be explained by the difference in assay conditions. We cannot explain precisely how *HRD3* suppression works *in vivo*, but the absence of M5 flipping activity (due to the absence of Rft1) in reconstituted proteoliposomes is not at odds with the phenotypes of *rft1Δ+HRD3*.

As discussed in the first revised manuscript or in the first rebuttal letter, the weak *in vivo* M5 flipping observed in *rft1Δ+HRD3* is the basis for the genetic suppression of *rft1Δ* lethality (Lines 192-193 of the 1st revised manuscript). This phenotype does not weaken but rather strengthen our conclusion that the essential function of Rft1 is M5-PP-Dol flipping.

Hrd3 is an ER membrane protein that functions as part of the ERAD-complex. ERAD regulates both protein and lipid homeostasis in the ER. It is conceivable that the combined overexpression of *HRD3* and deletion of *RFT1* alters the integrity of the ER membrane in a way that affects ER homeostasis. For instance, this could alter the substrate specificity of another flippase or scramblase. Our intention is to further examine the molecular mechanism for how this happens, but those studies are beyond the scope of the current work.

Our results have provided strong genetic and biochemical evidence demonstrating that **Rft1 is both necessary and sufficient for ER LLO translocation**. The experiments were performed rigorously, are well controlled and reproducible. Therefore, we do not agree that the title of our manuscript should be changed. We have added a paragraph in the “Discussion” for explaining our opinions (See Lines 265-294 in the second revision).

Point 3.

Reviewer’s comment:

Flippase assay kinetics (Figure 2).

As the authors state in the rebuttal document, the readout of the M5 flippase assay (Fig. 2) depends on a combination of flippase activity and mannosidase activity and suggest that the mannosidase step is rate-limiting. The half-time of the mannosidase step is 30 min, and this should be the same for the flippase-containing vesicles if the mannosidase is rate-limiting. For example, inspection of Fig 3 in ref 35 (Huang et al. 2021) shows that the rate of fluorescence reduction is the same in protein-free vesicles and in TMEM41B proteoliposomes in an assay where NBD-PC is the lipid (=M5 in the current assay) and dithionite is the probe (=mannosidase in this assay). I don't think that the analysis of the kinetics presented in the rebuttal document is correct - flipping seems to occur more slowly than the mannosidase reaction. This should be discussed properly in the text.

Response:

It appears we mis-read the original comments from the reviewer and improperly addressed their concern. We apologize for this mistake. Meanwhile, we thank the reviewer to point this out.

The mannosidase is rate-limiting. Theoretically, both empty liposomes and

proteoliposomes that contain protein without flippase activity can be used as negative controls. However, we observed a slight difference in M5 hydrolysis by α -1,2 mannosidase treatment of protein-free liposomes vs HhAgl22-containing proteoliposomes. As shown in Figure 2d, HhAgl22-reconstituted proteoliposomes displayed a similar half time of α -1,2 mannosidase treatment with flippase-containing proteoliposomes, while a slight delay was observed in protein-free liposomes. This result suggested the HhAgl22-containing proteoliposome is a better negative control than protein-free liposomes. It is possible that the reconstituted Agl22 protein, which has no flipping activity, slightly enhances the mannosidase activity *in vitro*. Nevertheless, the flipping activity of Rft1 is convincing. We added this point in the revision (see Lines 172-178 of the second revision)

Point 5. Technical issues.

Reviewer's comment:

(a) Line 486 - what mole percentage of M5 (or M3) lipid was used in the reconstitution?

Response:

5 nmol M3GN2-PP-Phy, M5GN2-PP-Phy or M5GN2-PP-Dol was used in the reconstitution. We have added this to the methods in the revision. (see Line 526 of the second revision)

Reviewer's comment:

(b) Line 488 - it seems that the authors dried lipids, dissolved in buffer with 1% TX-100 and then inexplicably extruded the detergent-solubilized lipids 30x though a 400-nm filter. Why was the extrusion done?

Response:

We appreciate the professional suggestion from reviewer. The treatment of extruding the detergent-solubilized lipids 30x though a 400-nm filter is unnecessary. The method used in this study strives to reconstitute liposomes with a diameter of approximately 200 nm (ref 23). Therefore, we have modified the relevant description in the "Methods". (see Lines 526-527 of the second revision)

Reviewer's comment:

(c) Protein and lipid recovery after reconstitution (provided in the rebuttal document) should be included in the paper.

Response:

Protein and lipid recovery were added to the manuscript in "Methods".

The phospholipid recovery rate of protein-free liposomes is $61.50 \pm 3.09\%$, while ScRft1 proteoliposomes revealed $59.54 \pm 1.73\%$ phospholipid recovery and $39.54 \pm 2.6\%$ protein recovery. Similar analyses for HhAgl22, HhAgl22 and HsRft1 proteoliposomes, the protein recovery was $55.10 \pm 2.5\%$, $60.04 \pm 4.41\%$ and $42.16 \pm 5.44\%$, and the corresponding phospholipid recovery rate were $64.19 \pm 0.98\%$, $61.30 \pm 1.17\%$ and $60.91 \pm 0.50\%$. (see Lines 535-542 of the second revision)

Reviewer's comment:

(d) Fig 3a - is this an FOA plate? how was the empty vector control set up?

Response:

It is a 5-FOA plate. Similar to the experiment shown in Fig 1b, the empty vector (YEp351-GAPII, see Table 2 for its description) was expressed in W303a-*rft1Δ* mutants containing pRS316-*ScRFT1*.

Reviewer's comment:

(e) Fig 3a - why was the growth curve done in YAPD media? why not YPD?

Response:

The W303a (*MATα ade2-1 ura3-1 his3-11 trp1-1 leu2-3,112 can1-100*) strain used in manuscript is prototrophic for adenine. It is common practice in the yeast genetics field to use YPAD (=YPD + 20 mg/L adenine) for growth of *ade2* mutants.

Reviewer's comment:

(f) Please include in Table 2 a description of the HRD3 plasmid used in Fig 3a.

Response:

The description of the *HRD3* plasmid has been supplemented in Table 2.

We added the following text in the legend of Figure 3.

--YEp351-GAPII plasmids, containing *ScRFT1*, *HRD3* or empty vector, were introduced in W303a-*rft1Δ* mutants containing pRS316-*ScRFT1*. Serial dilutions of each strain were plated on SD-Leu containing 5-FOA and incubated for 2 days at 30 °C -- (see Lines 350-353 of the second revision)

Reviewer's comment:

(g) Line 190 - the authors should add a quantitative comparison of the relative accumulation of M5 lipid in the HRD3 cells versus GALpr-Rft1 cells grown in glucose.

Response:

As described in our response to Point 2, our results presented in Figure 3 demonstrated that *rft1Δ+HRD3* cells do flip certain amount of M5-PP-Dol into the ER lumen and thus accumulate less M5 than GALpr-Rft1 cells grown in glucose. Thus, we directly described this phenotype in the result part. (see Lines 198-203 in second revision).

Reviewer's comment:

(h) The rebuttal data on M5Gn2-PP-Dol accumulation, and concomitant decrease in Gn2-PP-Dol, as a function of time in glucose media of the GALpr-RFT1 strain, should be included as supplementary data, accompanied by a graph showing the relative change), as this would help convince the reader of the quantitative quality of this readout.

Response:

Supplementary Fig. 1 has been added for showing the relative changes of M5 and M2

Supplementary Figure 1. Data supporting Fig. 1 | The inhibition of *RFT1* leads to the accumulation of M5GN2-PP-Dol

(a) UPLC chromatograms of oligosaccharides released from the LLO intermediates in *GALpr-RFT1* strain. The glucose (Glc)-repressible *GALpr-RFT1* strain grown in Glc for 0, 6 or 12 h at 30 °C, respectively. The LLO intermediates were extracted and the oligosaccharides were analyzed by UPLC-MS. Major peaks corresponding to GN2 and M5GN2 are marked. (b) The accumulation of M5 is positively correlated with the inhibition time of *RFT1*. The ratio is the peak area ratio in the UPLC chromatograms corresponding (a).

Reviewer's comment:

(i) The authors should state in the text that purified ScRft1 was verified by LC-MS/MS sequencing and show, as a supplementary figure, the sequencing data presented in the rebuttal.

Response:

Supplementary Fig. 5 has been added to confirm the identity of purified ScRft1 as ScRft1. The relevant sequencing data will be uploaded to the database as required.

Supplementary Figure 5. Data supporting Fig. 2 | LC-MS/MS analysis of purified ScRft1

(a) SDS-PAGE analysis of total proteins and purified FLAG-ScRft1 (top panel) and analyzed by Western blotting with anti-FLAG antibody (bottom panel). (b) LC-MS/MS chromatograms of peptides. The gel marked in the red box was cut out, the proteins were enzymatically hydrolyzed into peptides and analyzed by LC-MS/MS. (c) Matched peptides of Rft1 shown in red.

Reviewer's comment:

Of note, the original comment about the incorrect size came from the labeling of molecular weight markers in Fig 2c - in the original manuscript the protein ran between markers labeled 60 and 70 kDa; original manuscript, the gel remains the same, but the relevant markers are now 50 and 70 kDa. The gel figure (Fig 2c) has indeed been changed even though the authors say in their rebuttal that it remains the same.

Response:

The sizes of the protein molecular weight marker were corrected in revision. We thank the reviewer for noting it was incorrectly labeled in the original manuscript and apologize for the error.

Reviewer's comment:

(j) Line 233 - add a phrase: HRD3 lacked this activity, yet glycosylated CPY.

Response:

The translocation of M5 in *rft1Δ+HRD3* cells has been explained in more detail in the response letter and our revised manuscript (See Lines 265-294 in the second revision). Therefore, we believe the modification here is already not necessary.

Reviewer's comment:

(k) Line 269 - please add Plasmodium.

Response:

'*Plasmodium*' has been added.

Reviewer's comment:

(l) Line 65 - modify as follows: ... displayed robust LLO-specific flippase activity.

Response:

'... LLO-specific ...' has been added.

Reviewer #3 (Remarks to the Author):

Reviewer's comment:

I thank the authors for their effort to correct claims of having identified Rft1p. I may be sensitive but with the statements "The identity of this translocator is an important unanswered question" and "Genetic studies implicate Rft1" in the abstract, the authors are suggesting that Rft1 was not identified.

Response:

"The identity of this translocator is an important unanswered question" was removed from the "Abstract"; We've changed the "implicate" to "identified" (see the changes in "Abstract")

Reviewer #1 (Remarks to the Author):

I repeat certain points raised in the previous review(s) and emphasize others, as I don't find the rebuttal to be satisfactory. The bottom line is that the result of Fig 3 — on the basis of which the authors claim that Rft1 is necessary for LLO translocation — cannot be interpreted uniquely as the authors have done. I provide other possible, more reasonable, explanations.

1. The authors show that Rft1 and the unrelated protein HhAgl23 are able to flip M5-PP-Dol in an in vitro assay (HhAgl22 lacks activity, providing a control for their assay). This demonstrates sufficiency.

a. The in vitro assay has poor time resolution but is adequate for the purpose for which it is being used. The assay relies heavily on previous work which the authors cite liberally in the introduction, but it would be appropriate to refer to this previous work explicitly when describing their own assay, and to compare and contrast the methods (why choose a slow enzyme to probe M5-PP-Dol when the previously reported assay used a faster probe?).

b. The authors should also connect their results substantively with the previous in vitro data — for example, whereas Rft1 may be able to flip M5-PP-Dol as shown in the present paper, the previous data would suggest that there is at least another, much more abundant, flippase capable of doing the same thing. Phrased in this way, there is no conflict between these results. Indeed, the authors would help their case and advance the field if they were to repeat the previous data using their methods, i.e. reconstitute an extract of proteins from cells as in Fig. 3d WT and compare it with an extract from which Rft1 has been biochemically eliminated (use cells where the only copy of Rft1 carries an epitope tag to facilitate immunodepletion). This was suggested in the original round of review, but the authors seemed to be reluctant to carry out this simple test which would not require much work on their part and place the present paper on a good footing.

2. Issues remain concerning the experiment described in Figure 3 where the authors make use of a strain which lacks Rft1, but which is able to survive because of over-expression of the ubiquitin ligase Hrd3. I summarize (again) the key results as the authors have not addressed these concerns satisfactorily in their revision.

a. The rft1D-HRD3 cells grow almost normally (doubling time appears similar to that of wild-type cells, with a longer lag to get to exponential growth (Fig. 3a)) and can N-glycosylate CPY. This is important — even though glycosylation may not be as efficient as in WT cells, it nevertheless occurs, with a band corresponding to fully glycosylated CPY evident in Fig. 3b.

b. The rft1D-HRD3 cells accumulate M5-PP-Dol to a lesser extent than seen in true rft1 deficiency (Fig. 1d). The authors must provide numbers to indicate the absolute amounts of M5-PP-Dol in rft1D-HRD3 cells versus WT cells versus Glc-grown cells (Fig. 1d) - for example 1-unit M5-PP-Dol per OD600 WT cells, vs 100-units per OD600 Glc-grown cells, vs 20-units per OD600 rft1D-HRD3 cells. This issue was raised previously, see Technical Points — the authors claim to address this in lines 198-203, but no numbers are provided, and it is important to provide them. This is critical as the authors repeatedly resort to soft descriptions 'some, albeit very little' (line 282), 'residual translocation' (line 284), 'small amount of translocation' (lines 287-288) which are not helpful.

c. Thus, the cell-based data indicate that rft1D-HRD3 cells are able to grow and perform N-glycosylation, implying flipping of M5-PP-Dol. Based on their various statements in the text and in the rebuttal, the authors clearly agree with this assessment. For example, the authors write in their rebuttal to Point 2 "suppression of rft1Δ lethality is always accompanied by rescue of M5-PP-Dol flipping".

d. Thus, the authors present clear evidence of flipping in vivo in rft1D-HRD3 cells. However, they could not demonstrate flipping after reconstituting an extract from these cells. Their explanation is that a <2-fold increase in Hrd3 levels (Supp Fig 6c) perturbs the ER enabling flipping of M5-PP-Dol in cells. This apparently occurs without the participation of a specific protein(s)(lines 288-291) as they could not reconstitute the activity. I find this not very credible. With such a perturbation of ER structure they might expect to see global effects on the cell - is there any evidence for this? The authors also suggest that ER perturbation may affect the activity of other flippases that may substitute for Rft1. This seems more reasonable. However, why do the authors not see the activity of these other flippases in their

reconstituted assay? These are important issues as it is on the basis of these results and their interpretation of them that the authors claim that Rft1 is necessary for flipping M5-PP-Dol in cells. e. I have struggled to reconcile the disparity between the cell-based and in vitro data on the rft1D-HRD3 strain and can suggest the following explanations. I propose that the rft1D-HRD3 cells have an M5-PP-Dol flippase(s) (this is the same assessment presented by the authors, see point 2c above) but this protein(s) failed to be extracted/reconstituted efficiently and consequently its activity was not detected in the in vitro assay — (i) perhaps the protein is part of a complex that is not well-extracted in DDM (line 516) and therefore not present in the extract as used for reconstitution, (ii) the extract is prepared from membranes pelleted at 20,000xg, ignoring the true microsomal fraction which would need a harder spin (>150,000xg) — maybe the 'suppressor flippase' is in the microsomal fraction and not in the 20,000xg pellet?, and/or (iii) the flippase multimerizes during BioBead treatment (line 531) such that it reconstitutes into many fewer vesicles than expected, requiring PPRs in considerable excess of 20 in order to detect activity. This has been reported for other transport proteins, including lipid scramblases, in the literature. There may be other explanations, but at a minimum these options must be discussed, and the strength of the conclusions adjusted.

f. Because of these issues, the authors must scale back the title of their paper as previously suggested. They argue in their rebuttal that they have made their case in justifying the title, and that new text on lines 265-294 spells this out. I don't find this convincing for the reasons described in above and note again that the use of non-quantitative language (lines 281-284) to describe phenomena is misleading does not help. The authors should simply remove three words 'is necessary and' from their title to have a more accurate statement of their results.

3. Technical points:

a. Line 516: The authors resuspend the 20,000xg membrane pellet in lysis buffer containing DDM and then store at -80C. Is this accurate? Or was this preparation ultracentrifuged to remove insoluble material before storing at -80C? As written, this is a problem.

b. Line 527: the authors add buffer/salt containing 1% TX-100 to their dried lipids and vortex for 20 min. They state that this produces protein-free liposomes. This is incorrect — this will produce detergent/lipid micelles, not liposomes.

c. Quantitative aspects should be included in the paper — for example, (i) what can be learned about flipping kinetics and how it compares with previous reports (Fig. 2d), (ii) what is to be deduced from the plot in Fig. 2e which is labeled as 'translocation rate as a function of different protein to phospholipid ratios', and (iii) how do the HhAgl23-mediated kinetics of flipping of M3 compare with that of M5 (Fig. 4b vs Fig. 2d)?

4. Other points:

Lines 77-78: the results summary in the last part of the introduction only narrates data on sufficiency, i.e. Rft1 is sufficient for M5-PP-Dol flipping in vesicles. It is not possible to claim necessity based on this summary.

Lines 53-57, 291-292, 305 and other places in the text: the authors should make a serious attempt to reconcile their data and interpretation with previous in vitro data published a decade ago. They are using an identical reconstitution set-up as previously described, in which purified proteins or detergent extracts of yeast are reconstituted using the BioBeads method into vesicles. Their assay is conceptually similar to the flippase assay described previously. They report data with an extract of WT cells (Fig. 3d, WT) that would appear to be similar to previous data on WT extracts (except see point 2e(ii) and technical point 3a). They rely heavily on the previous data to set up the case, in the Introduction (lines 53-57), that there is a specific flippase activity for M5-PP-Dol. A proper explanation/reconciliation needs to be presented, considering the points raised above (items 1 and 2).

REVIEWERS' COMMENTS

Reviewer #1 (Remarks to the Author):

We thank the reviewer for their careful review of our manuscript and comments. We addressed the remaining points raised by the reviewer as much as possible and made revisions accordingly. Our changes, including the modifications and corrections requested from reviewers, are highlighted in the revised manuscript (emphasized with yellow background).

Reviewer's comment: (written in blue)

I repeat certain points raised in the previous review(s) and emphasize others, as I don't find the rebuttal to be satisfactory. The bottom line is that the result of Fig 3 — on the basis of which the authors claim that Rft1 is necessary for LLO translocation — cannot be interpreted uniquely as the authors have done. I provide other possible, more reasonable, explanations.

- 1. The authors show that Rft1 and the unrelated protein HhAgl23 are able to flip M5-PP-Dol in an in vitro assay (HhAgl22 lacks activity, providing a control for their assay). This demonstrates sufficiency.**

Response:

In deference to the reviewer and the editorial decision, we have removed “necessary” from the title and the manuscript, despite our disagreement with this idea. HhAgl23, a protein unrelated to Rft1, was used in our experiments not simply for demonstrating its biochemical *in vitro* M5GN2-PP-Dol flipping activity. We also demonstrated that this unrelated HhAlg23 M5GN2-PP-Dol flippase complements two inseparable *rft1Δ* phenotypes: inviability and N-glycosylation defects. Combined, these are powerful genetic and biochemical data that support our model that it is the **M5GN2-PP-Dol flipping activity of Rft1 that is essential for yeast cell growth**.

Reviewer's comment:

a. The in vitro assay has poor time resolution but is adequate for the purpose for which it is being used. The assay relies heavily on previous work which the authors cite liberally in the introduction, but it would be appropriate to refer to this previous work explicitly when describing their own assay, and to compare and contrast the methods (why choose a slow enzyme to probe M5-PP-Dol when the previously reported assay used a faster probe?).

Response:

- 1) We respect and recognize the important contributions of many scientists in this research field. Aebi and his group initially identified Rft1 as a flippase for the M5GN2-PP-Dol translocation. Rigaud and his group reconstituted proteoliposomes

and thus provided a tool for studying the activity of lipid flippases (*Biochimica et Biophysica Acta*, 1025 (1990) 179-190). Menon and his group developed the ConA capturing M5-PP-Dol assay system. While it is true that we and others benefited greatly from these prior studies, we disagree that our UPLC-MS-based M5GN2-PP-Dol flipping assay is “heavily” dependent on the previous ConA capturing M5-PP-Dol assay. They are completely different from one another. The only similarity is the method used for proteoliposome reconstitution, which in fact was first developed by Rigaud and other scientists.

- 2) Both the ConA-capture and UPLC-MS-based assays have unique advantages and disadvantages and as detailed in the 1st rebuttal letter, results from these two assays cannot be directly compared. Indeed, our assay confirmed that purified Rft catalyzes M5GN2-PP-Dol flipping while the ConA-capture assay did not observe a requirement for Rft1 in M5-flipping in membrane fractions extracted from yeast, including the *GAL* promoter-*RFT1* strain or the rat liver. Moreover, the putative Rft1-independent flippases purportedly assayed by the ConA-capture method nearly twenty years ago has yet to be identified. It is unreasonable to unilaterally request that we repeat those previous experiments simply to demonstrate which of the two assays are “better” beyond placating a reviewer who refuses to accept data that are both producible and of high-quality
- 3) Since development of the ConA-capture assay, no other flippase assay has been used and there has been little progress toward advancing studies of the ER flippase. The new assay system we describe will allow mechanistic studies of M5GN2-PP-Dol flipping and move the field forward. We believe that researchers through the future studies will decide whether our UPLC-MS-based assay can be accepted.
- 4) In this study, we focused determining whether or not Rft1 directly catalyzes M5GN2-PP-Dol flipping. It was not our intention to provide a comparative analysis of our M5GN2-PP-Dol assay system with the previous ConA-capture assay in this study. Those experiments are beyond the scope of this study, but we anticipate that the kinetic/enzymatic properties of Rft1 will be the focus of our next study.

Reviewer’s comment:

b. The authors should also connect their results substantively with the previous *in vitro* data — for example, whereas Rft1 may be able to flip M5-PP-Dol as shown in the present paper, the previous data would suggest that there is at least another, much more abundant, flippase capable of doing the same thing. Phrased in this way, there is no conflict between these results. Indeed, the authors would help their case and advance the field if they were to repeat the previous data using their methods, i.e. reconstitute an extract of proteins from cells as in Fig. 3d WT and compare it with an extract from which Rft1 has been biochemically eliminated (use cells where the only copy of Rft1 carries an epitope tag to facilitate immunodepletion). This was suggested in the original round of review, but the authors seemed to be reluctant to carry out this simple test which would not require much work on their part and place the present paper on a good footing.

Response:

- 1) We don't deny the possibility that another M5GN2-PP-Dol flippase may exist. However, our data refute the possibility that **this flippase is abundant or redundant with Rft1 in yeast**. The main reason is simple. If such abundant and or redundant flippases exist, they should also work in the *rft1Δ* strain and thus complement its growth lethality as seen by HhAgl23 *rft1Δ* complementation. As presented in the "DISCUSSION", the residual translocation of M5GN2-PP-Dol in the *HRD3+rft1Δ* strain may be explained by some type of compensatory mechanism triggered by the overexpression of *HRD3* in these *rft1Δ* cells. There is good evidence that Hrd3 forms a channel in the ER membrane that is used for retro-translocation of misfolded proteins and that this channel alters the ER membrane substantially (see Rappoport 2018). If such much more abundant flippases exist, they would fully complement the growth lethality and N-glycosylation of *rft1Δ* cells like the HhAgl23 does. Clearly, this not the case because *rft1Δ* cell is lethal.
- 2) About whether we should repeat the previous experiment using immunoprecipitation to remove the Rft1 protein, we have clearly detailed our opinions. We already performed experiments to test flippase activity using the extract from the *rft1Δ+HRD3* cells, in which the *RFT1* gene was completely removed. This is the better method of ensuring the absence of Rft1. Again, we are not in the position to judge those previous studies using our results, which have been done with different assay system and different protein samples. We should strive to provide reproducible high-quality data to interpret. We can promise that our experiments for testing the activity using the extract from the *rft1Δ+HRD3* cells have been carefully and professionally performed.

Reviewer's comment:

2. Issues remain concerning the experiment described in Figure 3 where the authors make use of a strain which lacks Rft1, but which is able to survive because of over-expression of the ubiquitin ligase Hrd3. I summarize (again) the key results as the authors have not addressed these concerns satisfactorily in their revision.

Response:

We thank very much to the reviewer for summarizing our results. However, we have different explanations in several points. We address them as follow:

Reviewer's comment:

a. *rft1Δ-HRD3* cells grow almost normally (doubling time appears similar to that of wild-type cells, with a longer lag to get to exponential growth (Fig. 3a)) and can N-glycosylate CPY. This is important — even though glycosylation may not be as efficient as in WT cells, it nevertheless occurs, with a band corresponding to fully glycosylated CPY evident in Fig. 3b.

Response:

The reviewer is suggesting that this strain is near wild type in its glycosylation capacity, but the opposite is true. We again emphasize the point that the CPY protein is hypo-N-glycosylated; most of it is not glycosylated

Reviewer's comment:

b. The *rft1D*-HRD3 cells accumulate M5-PP-Dol to a lesser extent than seen in true *rft1* deficiency (Fig. 1d). The authors must provide numbers to indicate the absolute amounts of M5-PP-Dol in *rft1D*-HRD3 cells versus WT cells versus Glc-grown cells (Fig. 1d) - for example 1-unit M5-PP-Dol per OD600 WT cells, vs 100-units per OD600 Glc-grown cells, vs 20-units per OD600 *rft1D*-HRD3 cells. This issue was raised previously, see Technical Points — the authors claim to address this in lines 198-203, but no numbers are provided, and it is important to provide them. This is critical as the authors repeatedly resort to soft descriptions 'some, albeit very little' (line 282), 'residual translocation' (line 284), 'small amount of translocation' (lines 287-288) which are not helpful.

Response:

We have added the Supplementary Figure 7 to the revised manuscript to quantitatively describe the different accumulation of M5GN2-PP-Dol in *rft1Δ+HRD3* cells and compared to that in the wild type or glucose-repressed *GALpr-RFT1* cells.

Reviewer's comment:

c. Thus, the cell-based data indicate that *rft1D*-HRD3 cells are able to grow and perform N-glycosylation, implying flipping of M5-PP-Dol. Based on their various statements in the text and in the rebuttal, the authors clearly agree with this assessment. For example, the authors write in their rebuttal to Point 2 "suppression of *rft1Δ* lethality is always accompanied by rescue of M5-PP-Dol flipping".

d. Thus, the authors present clear evidence of flipping in vivo in *rft1D*-HRD3 cells. However, they could not demonstrate flipping after reconstituting an extract from these cells. Their explanation is that a <2-fold increase in Hrd3 levels (Supp Fig 6c) perturbs the ER enabling flipping of M5-PP-Dol in cells. This apparently occurs without the participation of a specific protein(s)(lines 288-291) as they could not reconstitute the activity. I find this not very credible. With such a perturbation of ER structure they might expect to see global effects on the cell - is there any evidence for this? The authors also suggest that ER perturbation may affect the activity of other flippases that may substitute for Rft1. This seems more reasonable. However, why do the authors not see the activity of these other flippases in their reconstituted assay? These are important issues as it is on the basis of these results and their interpretation of them that the authors claim that Rft1 is necessary for flipping M5-PP-Dol in cells.

Response:

1) As we explained to the comments raised in point 1b (see above), based on our results, the possibility that another much more abundant M5GN2-PP-Dol flipping activity is low. The main reason is simple. If such flippases exist, they should also

work in the *rft1Δ* strain as well and then complement its growth lethality as is seen by HhAgl23 complementation. Our results cannot eliminate more complicated possibilities, for instance the activation of otherwise inactive M5GN2-PP-Dol flippases by *HRD3* overexpression. Thus, as requested by this reviewer and editor the word “necessary” was removed from the title and throughout the manuscript.

- 2) Our observation that *in vitro*, liposomes reconstituted with proteins extracted from this *HRD3+rft1Δ* strain failed to display any flippase activity, indicating no detectable M5GN2-PP-Dol flippase activity involved in the membrane extract. This result supports our explanation that *in vivo*, the observed residual translocation of M5GN2-PP-Dol may be due to an alteration of the integrity, structure, or protein abundance of the ER membrane (we have clearly presented this issue in the DISCUSSION of our submitted manuscripts. See Lines 276-294 in 2nd revised manuscript). The possibility of existing another M5GN2-PP-Dol flipping activity is low.

Reviewer’s comment:

e. I have struggled to reconcile the disparity between the cell-based and *in vitro* data on the *rft1D-HRD3* strain and can suggest the following explanations. I propose that the *rft1D-HRD3* cells have an M5-PP-Dol flippase(s) (this is the same assessment presented by the authors, see point 2c above) but this protein(s) failed to be extracted/reconstituted efficiently and consequently its activity was not detected in the *in vitro* assay — (i) perhaps the protein is part of a complex that is not well-extracted in DDM (line 516) and therefore not present in the extract as used for reconstitution, (ii) the extract is prepared from membranes pelleted at 20,000xg, ignoring the true microsomal fraction which would need a harder spin (>150,000xg) — maybe the 'suppressor flippase' is in the microsomal fraction and not in the 20,000xg pellet?, and/or (iii) the flippase multimerizes during BioBead treatment (line 531) such that it reconstitutes into many fewer vesicles than expected, requiring PPRs in considerable excess of 20 in order to detect activity. This has been reported for other transport proteins, including lipid scramblases, in the literature. There may be other explanations, but at a minimum these options must be discussed, and the strength of the conclusions adjusted.

Response:

- 1) We think our opinion in point 2c has been replaced here. Our results do reveal the M5GN2-PP-Dol flipping in the *rft1Δ+HRD3* strain, but we didn’t confirm and explain that this translocation comes from a flippase.
- 2) We thank the reviewer’s advice for reconciling the disparity between the cell-based and *in vitro* data on the M5GN2-PP-Dol flipping activity in *rft1Δ+HRD3* strain. We performed this assay in parallel with the membrane extract from the wild type cells, which M5GN2-PP-Dol activity has been clearly detected. Failure to extract this activity should therefore not be the reason to suspect the authenticity of our results. We performed experiments rigorously and carefully. Thus, the M5GN2-PP-Dol flipping observed in *rft1Δ+HRD3* cells is not catalyzed by another flippases but

caused by the ER perturbation.

Reviewer's comment:

f. Because of these issues, the authors must scale back the title of their paper as previously suggested. They argue in their rebuttal that they have made their case in justifying the title, and that new text on lines 265-294 spells this out. I don't find this convincing for the reasons described in above and note again that the use of non-quantitative language (lines 281-284) to describe phenomena is misleading does not help. The authors should simply remove three words 'is necessary and' from their title to have a more accurate statement of their results.

Response:

We accept this suggestion and have made the changes.

3. Technical points:

a. Line 516: The authors resuspend the 20,000xg membrane pellet in lysis buffer containing DDM and then store at -80C. Is this accurate? Or was this preparation ultracentrifuged to remove insoluble material before storing at -80C? As written, this is a problem.

Response:

This method mainly extracts ER membrane protein components. The membrane fraction was resuspended into 5 mL lysis buffer with 1% DDM and directly used to construct liposomes. for detecting the activity of flipping enzymes, as shown in Fig. 3d. We have made corrections in the corresponding methods section.

b. Line 527: the authors add buffer/salt containing 1% TX-100 to their dried lipids and vortex for 20 min. They state that this produces protein-free liposomes. This is incorrect — this will produce detergent/lipid micelles, not liposomes.

Response:

We thank the reviewer for pointing this out and we agree with this comment. This step will produce detergent/lipid micelles, not liposomes. Subsequently, liposomes will gradually form with the addition of Bio-beads. We have made corrections in the corresponding methods section.

c. Quantitative aspects should be included in the paper — for example, (i) what can be learned about flipping kinetics and how it compares with previous reports (Fig. 2d), (ii) what is to be deduced from the plot in Fig. 2e which is labeled as 'translocation rate as a function of different protein to phospholipid ratios', and (iii) how do the HhAg123-mediated kinetics of flipping of M3 compare with that of M5 (Fig. 4b vs Fig. 2d)?

Response:

1) The flipping enzyme activity detection system in this work is based on the dual

enzyme method, and This is not suitable for flipping kinetic quantification of RFT1. We verified that the purified Rft1 exhibits the activity of flipping M5, although the detection time is relatively long (Fig. 2d).

- 2) The result in Fig. 2e demonstrated the increased M5GN2 to M3GN2 conversion as a function of protein/phospholipid ratio and the M5GN2-PP-Phy flipping by Rft1 is dose-dependent.
- 3) The mannosidases used in Fig 4b (α 1-2 mannosidase) and Fig 2d (α 1-2,3 mannosidase) are different. Therefore, it is inappropriate to compare the HhAgl23-mediated kinetics of flipping of M3 compare with that of M5. Our results (Fig. 4b and Fig. 2d) are enough to demonstrate that HhAgl23 exhibits the activity of flipping M3 and M5.

4. Other points:

Lines 77-78: the results summary in the last part of the introduction only narrates data on sufficiency, i.e. Rft1 is sufficient for M5-PP-Dol flipping in vesicles. It is not possible to claim necessity based on this summary.

Response:

We have made the changes.

Lines 53-57, 291-292, 305 and other places in the text: the authors should make a serious attempt to reconcile their data and interpretation with previous in vitro data published a decade ago. They are using an identical reconstitution set-up as previously described, in which purified proteins or detergent extracts of yeast are reconstituted using the BioBeads method into vesicles. Their assay is conceptually similar to the flippase assay described previously. They report data with an extract of WT cells (Fig. 3d, WT) that would appear to be similar to previous data on WT extracts (except see point 2e(ii) and technical point 3a). They rely heavily on the previous data to set up the case, in the Introduction (lines 53-57), that there is a specific flippase activity for M5-PP-Dol. A proper explanation/reconciliation needs to be presented, considering the points raised above (items 1 and 2).

Response:

We have made our comments at the point 1a.